# CELLDJBENCH: BENCHMARK DATASETS FOR DATA-DRIVEN BIOLOGICAL FLUID SIMULATION

## ABSTRACT

Biological fluid simulation is a critical tool for comprehending the intricate and complex fluid dynamics that occur within bio logical systems. Recently, data-driven techniques have emerged as a promising avenue to enhance the accuracy and efficiency of biological fluid simulations. However, the community encounters two challenges. (1) Existing biological datasets only capture static snapshots, lacking the ability to capture dynamic biological processes. (2) These datasets are limited in scale due to the demanding experimental conditions. To address these challenges, this paper introduces four comprehensive large-scale datasets: Tension, Wets, CellDivision and Jellyfish, containing a wealth of biological dynamics and pushing the boundary of data-driven methods. These datasets have been meticulously designed to encompass a wide array of biological fluid dynamics scenarios. By incorporating physical modeling techniques such as phase-field method, these datasets provide a standardized evaluation framework for data-driven approaches. They empower researchers to objectively assess and compare different methodologies, fostering advancements in the field of biological fluid simulation. Furthermore, the availability of these benchmark datasets facilitates reproducibility and enhances the comparability of results across studies, promoting knowledge sharing and collaboration within the research community. Researchers can build upon existing models, leading to cumulative progress in the development of accurate and efficient data-driven models for simulating complex fluid dynamics within biological systems. We offer benchmark code and CellDJBench dataset link through the following link: https://anonymous.4open.science/r/ CellDJBench.

## 1 INTRODUCTION

Biological fluid simulation (Liu, 2005; Marques et al., 2011) plays a crucial role in various scientific and engineering domains, enabling the investigation of complex fluid dynamics (Tian et al., 2014) within biological systems. The accurate modeling of fluid behavior is essential for understanding phenomena such as blood flow in cardiovascular studies, the motion of swimming organisms, and the transport of nutrients within biological tissues. Traditional approaches (Kucera et al., 2024; Peng et al., 1999) to fluid simulation, have been widely used but often struggle to capture the intricate and complex dynamics exhibited by biological fluids. In recent years, data-driven approaches (Chen et al., 2016a; Weng & Zhu, 2021), leveraging machine learning (O'shea & Nash, 2015) and deep learning techniques (Ho et al., 2020), have shown great promise in enhancing the accuracy and efficiency of biological fluid simulations.

However, the adoption of data-driven techniques like FNO (Li et al., 2020) and SFNO (Bonev et al., 2023) in biological fluid simulation is hindered by several challenges. Importantly, the lack of standardized large-scale datasets (Hassan et al., 2023) poses a significant obstacle to evaluating and comparing different methodologies. Existing datasets are often limited in scope (Ahlmann-Eltze & Huber, 2023; Yang et al., 2022), failing to capture the diverse range of biological fluid dynamics or encompassing challenging scenarios. This limitation inhibits the objective assessment of various approaches and hampers the reproducibility and comparability of research findings. Developing models that can effectively handle the diverse physical properties, boundary conditions, and dynamic behavior of biological fluids remains an active area of research.

Although researchers have explored a lot on constructing biological datasets, there are some challenges that existing biological datasets cannot solve. (1) Existing biological datasets only capture biological snapshots, which cannot capture biological dynamics. (2) Current datasets rely on rigorous experimental conditions, such as specific temperature settings, making them challenging to acquire and limited in scale. Some experiments, like scRNA-seq on cell, are destructive, thereby hindering observations of cell trajectories. In contrast, living-image (in video format) data demonstrates exceptional congruence with physical modeling techniques such as phase-field methods. These models harness the quantitative intricacy inherent in live-imaging data to establish computational representations of cellular behavior, providing a robust framework for theoretical investigation and hypothesis validation. Thus, to solve these challenges, we aim to construct large-scale biological living-image datasets like single-cell dynamic datasets via phase-field methods that are beneficial for capturing biological dynamics and further providing a valuable resource to the research community.

In this paper, we create and release four large-scale biological datasets: two cell evolving datasets (i.e., Tension and Wets), one cell division dataset (CellDivision) and one Jellyfish dataset. These datasets are specifically designed to capture a wide range of biological fluid dynamics scenarios, encompassing tension-driven flows, wetting phenomena, cell divisions, and the intricate motion of jellyfish-like organisms. Each dataset is meticulously curated, incorporating variations in physical parameters, such as adhesion, tension, and flow rates, to simulate the complexity and heterogeneity of real-world biological systems.

Meanwhile, we also provide standardized evaluation benchmarks for data-driven biological fluid simulation, enabling researchers to benchmark their models against challenging and diverse scenarios. By establishing a common ground for comparison, these datasets facilitate the objective assessment and comparison of different data-driven methodologies, fostering the advancement of the field. Furthermore, the availability of these benchmark datasets promotes reproducibility and comparability of results across studies, highlighting the strengths and limitations of each approach in capturing the intricate dynamics of biological fluids. Researchers can build upon and improve existing models (Schindler & Kutzelnigg, 1983; Ronneberger et al., 2015), enabling cumulative progress in biological fluid simulation. In summary, our contributions are provided as follows:

- **New Perspective.** We are the first to investigate fluid dynamics modeling in biological systems at both the dynamic single-cell level and the jellyfish level. Unlike previous single-cell research focusing on a single snapshot, we simulate data across different time points, pioneering future research such as cell division simulation.
- **Large-scale Biological Datasets**. We collect around 1.3 TB of raw data using a new biological technique, resulting in four large-scale fluid dynamics datasets including Tension, Wets, CellDivision and Jellyfish. Our datasets have been made publicly available.
- **Extensive Benchmarks**. We make the first attempt to adopt 11 data-driven approaches for our new dynamics modeling problem, validating the potential applications of machine learning in single-cell biological systems. Our benchmark helps researchers evaluate their data-driven approaches to advance the field.

## 2 CELLDJBENCH: PRELIMINARY AND PROBLEM SETUP

In this section, we aim to provide preliminary information on the phase-field model and Lily-Pad simulator, which are shown in Figure 1. Then we provide the problem definition.

### 2.1 MOTIVATION TO CONSTRUCT BIOLOGICAL DYNAMIC DATA

Unraveling the intricacies of cellular and multicellular dynamics is paramount in current biological research. Two primary approaches have emerged for this purpose, each offering distinct strengths and limitations. Single-cell RNA sequencing (scRNA-seq) (Kolodziejczyk et al., 2015; Hwang et al., 2018) represents a powerful snapshot method, providing a population-level view of cellular states through gene expression analysis. However, scRNA-seq relies on cell destruction, hindering direct observation of individual cell trajectories. To address this limitation, researchers have developed single-cell trajectory inference techniques (Saelens et al., 2019; Cannoodt et al., 2016). While valuable for identifying population trends, these methods are inherently inferential, relying on complex algorithms susceptible to user bias and data quality. Live-imaging, in contrast, offers a compelling alternative. By directly visualizing individual cells over time, live-imaging captures the

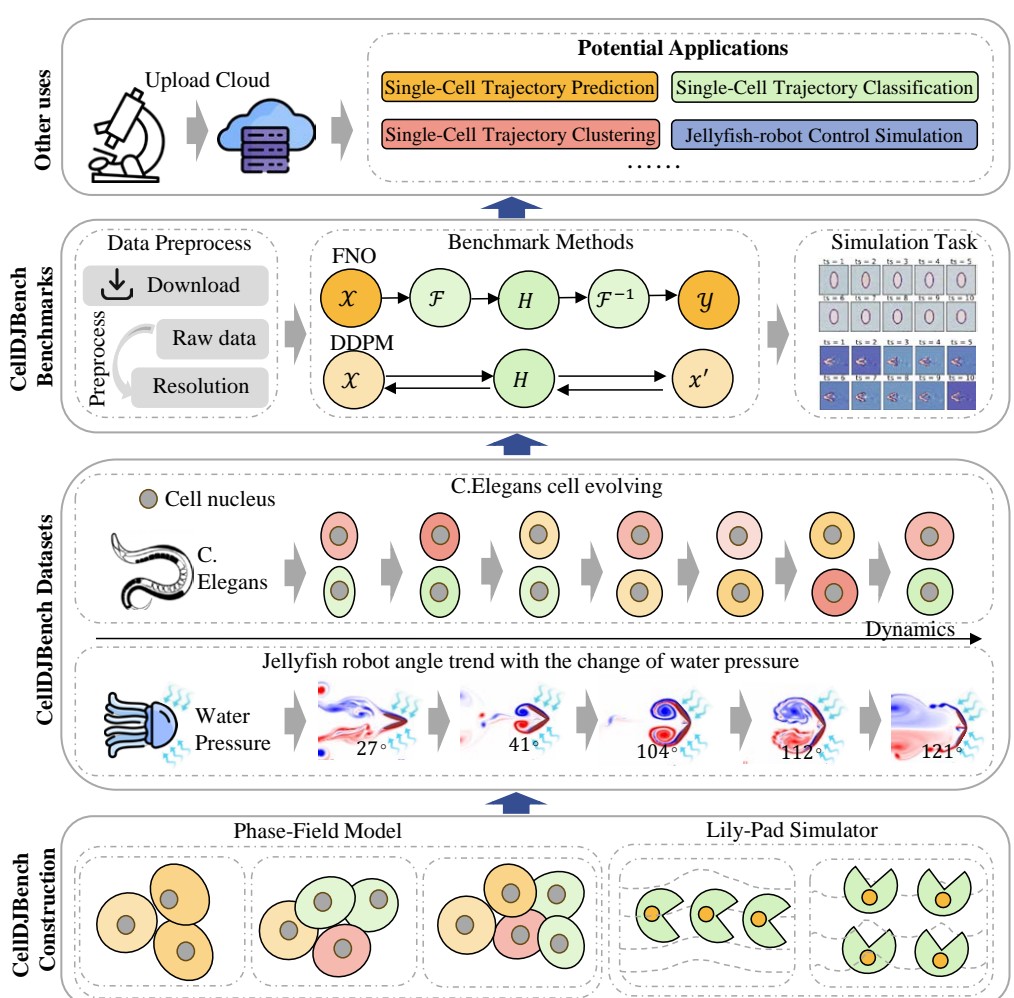

Figure 1: CellDJBench: It comprises four key components including construction in Section 2, CellDJBench datasets in Section 3, CellDJBench benchmarks in Section 4 and other potential uses in Section 3. Each section provides comprehensive and detailed descriptions of the respective component, ensuring a thorough understanding of each aspect.

dynamic choreography of cellular behavior with unparalleled fidelity. This approach transcends the limitations of scRNA-seq by enabling continuous monitoring of the same cell, revealing its unique response to stimuli and its progression through developmental pathways. Furthermore, live-imaging data exhibits remarkable compatibility with physical modeling techniques like phase-field methods. These models leverage the quantitative richness of live-imaging data to construct in silico representations of cellular behavior, offering a powerful framework for theoretical exploration and hypothesis testing.

## 2.2 How to Obtain Biological Dynamic Data?

**Phase Field Model:** The phase-field model, whose main idea and formulation are shown in (Alert & Trepat, 2020), which is a method that accounts for diffuse interfaces, has demonstrated its ability to reconstruct the embryonic morphologies of C. Elegans in previous studies (Jiang et al., 2019; Kuang et al., 2023; 2022). In this particular model, each cell is represented by a scalar order parameter $\phi_i(r)$, where $\phi_i$ ranges from 0 (indicating the cell is outside) to 1 (indicating the cell is inside). The variable i denotes the cell identity, and $r \in \mathbb{R}^3$ represents the three-dimensional spatial variable. The behavior of the cell is governed by various factors, including volume constriction, surface tension, repulsion from the eggshell, as well as repulsion and attraction from neighboring cells (Jiang et al., 2019; Kuang et al., 2023). The eggshell itself is simplified as an ellipsoid, with its three axes aligned parallel to the $x$ (anterior-posterior, A-P), $y$ (left-right, L-R), and $z$ (dorsal-ventral, D-V) axes of the

rectangular coordinate system. To mimic the lateral compression observed during in vivo imaging, the regions $|y| > w_{max}$ are truncated (Cao et al., 2020). The phase-field function describing the eggshell is expressed as $\phi_e = \frac{1+\tanh(\frac{d(x,y,z)}{\sqrt{2}\epsilon})}{2}$. The boundary width of the eggshell is controlled by the parameter $\epsilon = \frac{\sqrt{2}}{20}$. The distance of a point $(x, y, z)$ in space to the eggshell is denoted as $d(x, y, z)$ and shown as follows:

$$d(x,y,z) = \min\left\{\sqrt{x^2 + \frac{L_{x/A-P}^2}{L_{y/L-R}^2}y^2 - \frac{L_{x/A-P}^2}{L_{z/D-V}^2}z^2} - L_{A-P}, |y| - w_{max}\right\} \tag{1}$$

The dimensions of the ellipsoid are determined as $L_{x/A-P} = 27.7846\mu m$, $L_{y/L-P} = 14.7022\mu m$, and $L_{z/D-V} = 18.3778\mu m$, representing the lengths of its semimajor axes. The maximum width of the ellipsoid is denoted as $w_{max} = 10.1188\mu m$. These values are derived from experimental data. Additionally, the terms $\phi_e = 1$ and $\phi_e = 0$ correspond to the external and internal regions of the eggshell, respectively. To maintain a constant cell volume over time, the relative error between the simulated and experimental volumes is restricted, which is expressed as $\mathbf{F}_{\text{volume}} = \mathbf{M}[\frac{\int_\Omega \phi_i d\mathbf{r}}{V_i(t)} - 1]\mathbf{n}$. The volume constriction strength is denoted by $\mathbf{M} = 8$. The designed volume at a specific time point $t$ is represented by $V_i(t)$. The unit vector $\mathbf{n}$ is perpendicular to the cell boundary, directing inward toward the cell. A repulsive force acts between the cells and the eggshell, restricting their interaction. This repulsive force ensures that phase fields of cells and eggshells do not overlap, following:

$$\mathbf{F}_{\text{repulsion}} = \left(g\phi_i \sum_{j\neq i}^{N} \phi_j^2 + g_e\phi_i\phi_e^2\right)\frac{\nabla\phi_i}{|\nabla\phi_i|^2}. \tag{2}$$

And $g = 1.6$ and $g_e = 16$ are the repulsive force between cells and the eggshell. Additionally, two additional significant mechanical forces are incorporated to characterize cellular and multicellular mechanics. The attractive forces between two cells are described as $\mathbf{F}_{\text{adhesion}} = \sum_{j\neq i}^{N} \alpha_{i,j}\nabla\phi_j$. The adhesion strength between cell $i$ and $j$ is represented by $\alpha_{i,j}$. The gradient operator is denoted by $\nabla$. The cell's stiffness is characterized by its surface tension, which influences the cell to adopt a spherical shape with reduced surface area and minimal deformation.

$$\mathbf{F}_{\text{tension}} = -\gamma_i\left[\triangle\phi_i - cW'(\phi_i)\right]\frac{\nabla\phi_i}{|\nabla\phi_i|^2} \tag{3}$$

The surface tension of cell $i$ is given by $\gamma_i$. The boundary width of the cell is controlled by the parameter $c = 1$. The function $W(\phi_i)$ transforms $\phi_i$ to a binary form, taking values of 0 and 1. Specifically, $W'(\phi_i) = 2\phi_i(\phi_i - 1)(2\phi_i - 1)$. The Laplacian operator is denoted by $\triangle$.

**Lily-Pad Simulator:** This is a powerful tool for simulating the complex fluid-structure interactions (Weymouth, 2015). Its primary aim is to make CFD more accessible by employing straightforward, high-speed methods that provide instant visual feedback to users. To truly fulfill this objective, Lily-Pad operates as a genuine CFD solver by solving the full two-dimensional Navier-Stokes equations and applying precise body boundary conditions. It simplifies many of the complexities commonly associated with CFD through the use of the Boundary Data Immersion Method (BDIM) (Weymouth & Yue, 2010). This technique immerses solid bodies into the fluid domain, streamlining the setup and execution of simulations. Despite its simplicity, BDIM offers high accuracy, possesses beneficial analytical properties, and has undergone extensive validation.

## 2.3 DATA-DRIVEN BIOLOGICAL SIMULATION PROBLEM SETUP

In this section, we aim to provide the problem definition of the data-driven biological dataset simulation. **Input:** biological features $\mathcal{X} = \{x_1, x_2, ..., x_n\}$, $x_i \in \mathbb{R}^{T\times D\times D}$ which are built upon biological datasets, namely cell evolving datasets and the jellyfish dataset. **Output:** the task is to learn a simulation function $f : x_i \to y_i$ that aims to simulate $x_i$ and obtain the simulated $y_i$ via optimizing data-driven methods. And the process is defined via the following equation:

$$\hat{f} = \arg\min_f \sum_1^n \sum_{t=1}^T \mathcal{L}(y_i, x_i) \tag{4}$$

Table 1: Overview of Jellyfish and Cell Datasets

| Datasets | Dimension | Type-Physics | Channels | Resolution | Times steps | Size | Videos | Images |
|---|---|---|---|---|---|---|---|---|
| Tension | 3D | Cell evolving | 1 | $256 \times 256$ | 20 | ~500G | 60,000 | 1,200,000 |
| Wets | 3D | Cell evolving | 1 | $256 \times 256$ | 20 | ~500G | 60,000 | 1,200,000 |
| CellDivision | 4D | Cell evolving | 1 | $60 \times 120 \times 80$ | 7-20 | ~320G | 300 | 4,000 |
| Jellyfish | 3D | Jellyfish-Robot | 3 | $256 \times 256$ | 20 | ~10G | 500 | 10,000 |
| Summary | - | - | - | - | - | ~1.3T | **120,800** | **2,414,000** |

where $T$ represents the number of time steps and $D$ is the dimension. $\mathcal{L}$ represents the loss function and denotes the Mean Squared Error (MSE) loss in this paper. And $n$ is the number of samples in each biological dataset.

## 3 CELLDJBENCH DATASETS CONSTRUCTION

In this section, we aim to present visualizations of the CellDJBench datasets, as depicted in Figure 1. Firstly, we highlight the rationale behind constructing biological datasets. Subsequently, we provide comprehensive explanations of the Tension, Wets, CellDivision and Jellyfish dataset constructions, along with their respective specifications and extensions. We also provide statistics of CellDJBench datasets in Table 1.

### 3.1 TENSION CONSTRUCTION

The purpose of the dataset is to examine the evolution of the system's shape under specific grid and domain conditions, achieved by adjusting parameters like tension parameters. A significant aspect of the simulation revolves around calculating the changes in the variable $\phi$ at each time step through the solution of a partial differential equation. As the simulation progresses, the values of $\phi$ are systematically updated across the entire two-dimensional spatial grid. A detailed construction process is shown in Appendix A.3.

The Tension dataset, focusing on cell evolving in a 3D environment, plays a crucial role in advancing our understanding of cellular biology. Cell evolving is a fundamental process in living organisms, and studying its dynamics and mechanisms has numerous applications in various fields, including medicine, genetics, and developmental biology. This dataset provides a valuable resource for researchers and scientists to explore the intricate details of cell evolving. With a resolution of 256x256 pixels and 20 time steps, it captures the progression of cell evolution, allowing for the analysis of morphological changes, cellular behavior, and the influence of external factors. The Tension dataset includes 2,500 videos and 50,000 images, offering a rich source of visual information. Researchers can use this dataset to develop and evaluate algorithms, models, and techniques for cell evolving analysis, identification of abnormal cell behavior, and understanding cellular processes. The availability of this dataset empowers the scientific community to make significant advancements in cellular biology, contributing to advancements in medicine, genetics, and various other disciplines.

**Specifications**. In contrast to existing single-cell datasets like (Ahlmann-Eltze & Huber, 2023), our dataset offers a unique perspective by capturing the dynamic characteristics of cell evolution through the measurement of tension changes in the tension dataset. This is achieved by analyzing single-cell living-images, allowing us to observe the relationship between cell shape and tension fluctuations. Additionally, we record the trajectory changes of individual cells, which holds significant potential for advancing research in areas such as single-cell trajectory classification and clustering.

Moreover, our tension dataset is distinguished by its large-scale nature, providing a robust foundation for future investigations utilizing large language models or other foundational models. This extensive dataset opens up possibilities for novel research inquiries and empowers the scientific community to explore new avenues in the study of cellular dynamics and behavior.

**Extensions**. The utilization of our tension dataset extends beyond its immediate application in capturing tension changes. It can also be leveraged for other single-cell tasks, such as single-cell trajectory prediction and single-cell trajectory clustering. Furthermore, our tension dataset stands out due to its high-resolution nature, with each snapshot providing a resolution of $256 \times 256$. This

high-resolution attribute introduces potential advantages for tasks that require detailed and precise analysis, opening up possibilities for exploring high-resolution applications in the future.

## 3.2 WETS CONSTRUCTION

The Wetting dataset aims to explore the changes in morphology of a system under various conditions, including grid and domain configurations, tension, adhesion, and different physical parameters. This dataset is generated through simulations that involve systematic modifications of key parameters to observe and analyze the wetting behavior that emerges as a result. By providing a comprehensive collection of simulated data, the Wetting dataset enables researchers to investigate the intricacies of wetting phenomena and gain insights into the underlying mechanisms governing wetting behavior in different scenarios. For a comprehensive understanding of the construction process, please refer to Appendix A.3, where a detailed description of the step-by-step construction process is provided.

**Specifications**. Our wets dataset offers a distinct perspective by capturing the dynamic characteristics of cell evolution through the assessment of wetting behavior. By analyzing single-cell living-images, we are able to delve into the intricate relationship between cell shape and wetting behavior, specifically by observing changes in adhesion. In addition to documenting trajectory changes of individual cells, this dataset holds substantial potential for advancing research in areas like single-cell trajectory prediction and clustering.

Furthermore, our wets dataset stands out due to its extensive scale, serving as a robust foundation for future investigations that employ large language models or other foundational models. The breadth and depth of this dataset unlock opportunities for novel research inquiries, empowering the scientific community to explore fresh avenues in the study of cellular dynamics and wetting behavior.

**Extensions**. Wets dataset goes beyond its initial objective of capturing dynamic wetting behavior, as it can be effectively utilized for various other single-cell tasks, such as single-cell trajectory inference and single-cell trajectory metabolomics. Furthermore, the high-resolution nature of our wets dataset, with each snapshot offering a resolution of $256 \times 256$, presents significant advantages for tasks that require detailed and precise analysis. This high-resolution attribute opens up possibilities for exploring applications that rely on meticulous examination and examination of cellular dynamics.

## 3.3 CELLDIVISION CONSTRUCTION

Our CellDivision dataset was created utilizing MATLAB software, employing numerical simulations to model cell growth and division processes during Natural condition. The study revolves around the generation of stochastic cell characteristics, with the progression of cell life cycles simulated within a three-dimensional space using the Finite Difference Method (FDM). Various aspects of cell dynamics such as division events, mechanical properties, trajectory paths, and velocity estimations are modeled. The simulations are executed on a 3D grid with a spatial resolution of $dx = 0.5$ in the spatial domain, while the temporal evolution is captured with a time step of $dt = 1$. The grid size and time step configuration are optimized to ensure precise representation of subtle cellular movements and state transitions. Each simulation iteration yields randomized values for cell division axes, volumes, and lifespans. Parameters $\sigma = 0$ and $c = 2$ regulate the mechanical characteristics of the simulated cells, enabling the investigation of cell behaviors under diverse environmental conditions.

The CellDivision dataset encompasses detailed cell attributes, time-series records, and trajectory paths. Converting this data into CSV format facilitates further analysis and modeling procedures. This dataset offers high-resolution temporal data sequences and spatial distribution details, which are invaluable for in-depth exploration of cellular dynamics and related phenomena.

**Specifications**. Our CellDivision dataset offers an unprecedented opportunity to advance our understanding of cell division dynamics under natural conditions. To be specific, it incorporates crucial biological details, including complete cell lineages (*e.g.*, AB, E, MS, P), precisely determined cell fates (*e.g.*, Muscle, Intestine), and the environmental conditions (*e.g.*, Natural) influencing development. Researchers can now explore the intricate, dynamic trajectories of individual cells as they evolve, providing insights into the complex interplay of factors governing cell division. The high-resolution nature of the data, achieved through sophisticated numerical simulation, mirrors the detail achievable in state-of-the-art experimental techniques. This remarkable similarity to experimentally

derived datasets makes it an ideal tool for validating new analytical methods, testing hypotheses about cell division mechanisms, and developing predictive models of cell behaviors.

**Extensions**. This dataset's potential extends beyond its immediate application. Future expansions could incorporate additional cell types, variations in environmental parameters (*e.g.*, nutrient levels, signaling molecule concentrations), and the introduction of cell-cell interactions. Integrating gene expression data would provide a multi-omics perspective, linking genotype to phenotype. Furthermore, exploring different simulation parameters ($\sigma$ and $c$) could uncover novel insights into the mechanical regulation of cell division. These extensions would create a powerful, versatile resource for the broader biological research community.

### 3.4 JELLYFISH CONSTRUCTION

Using the Lily-Pad simulator (Weymouth, 2015), we generate training and testing datasets. A $128 \times 128$ grid represents the 2D flow field, extending infinitely. The jellyfish's head is fixed at $(25.6, 64)$, while its symmetric wings are ellipses with 20 boundary points. Wing shape maintains a fixed 0.15 ratio between shorter and longer axes. The control signal, denoted as $w$, determines wing opening angle. Trajectories follow a periodic cosine curve with a period of $T' = 200$, starting with the widest angle. Simulated trajectories consist of 600 steps, stored as a sequence of 20 discrete steps from $T' = 200$ to $3T' = 600$. The datasets include PDE states, opening angles, boundary points, and image-like representations of boundaries. The Jellyfish dataset captures dynamic jellyfish locomotion in a 3D environment using a Jellyfish-Robot. It comprises 500 videos and 10,000 images, providing a comprehensive resource for studying aquatic organism locomotion and driving advancements in biomimetics, robotics, and marine biology. Further details can be found in Appendix A.3.

**Specifications**. Our Jellyfish dataset provides a unique perspective on the dynamic characteristics of the relationship between jellyfish-robot angle and water pressure. By analyzing living-images of jellyfish-robot angle change trends, we gain valuable insights into this intricate relationship. Moreover, this dataset not only captures trajectory changes of individual jellyfish but also holds significant potential for advancing research in underwater robot design and control. Additionally, our dataset stands out in terms of its extensive scale, allowing researchers to leverage large language models or other foundational models. The comprehensiveness and depth of this dataset open up new avenues for innovative research inquiries, empowering the scientific community to explore novel aspects of jellyfish-robot dynamics.

**Extensions**. The Jellyfish dataset continues to make a significant impact on the scientific community. It offers valuable insights applicable to various domains, including the design and control of underwater autonomous vehicles (AUVs), commonly known as underwater robots. The dataset's extensive scope and high-resolution attributes have the potential to enhance the applications of the Jellyfish dataset within the community.

## 4 CELLDJBENCH BENCHMARKS

In this section, our goal is to provide detailed descriptions of benchmark methods applied to the CellDJBench datasets, which are also illustrated in Figure 1. First, we offer concise explanations of the benchmark methods employed. Subsequently, we present the results obtained from applying these benchmark methods, along with a thorough analysis and interpretation of those results.

**Benchmark Methods:** In this section, we introduce baselines for simulating evolving cells: FNO (Li et al., 2020), SFNO (Bonev et al., 2023), TFNO, UNet (Ronneberger et al., 2015), and DDPM (Ho et al., 2020). FNO utilizes Fourier space for resolution-invariant global convolutions but has high memory requirements. SFNO uses spherical harmonics for handling spherical data with optimized memory usage. TFNO is a variant of FNO that adopts Tucker factorization of the weights. UNet is renowned for precise object localization and segmentation, achieving exceptional performance in various applications. DDPM gradually transforms noise-corrupted data to the original distribution through iterative diffusion, with training involving sampling and loss function computation. ConvLSTM (Shi et al., 2015) proposes the prediction of future precipitation patterns as a task of forecasting spatiotemporal sequences. ViT (Dosovitskiy et al., 2020) proposes the dependence on CNNs is not essential and transformer applied directly to sequences of image patches can achieve excellent performance in classifying image. MLP-Mixer (Tolstikhin et al., 2021) is solely built upon multi-layer

Table 2: Comparison of benchmarks in terms of MSE and Relative L2 norm on two datasets.

| Time Step | Metrics | FNO | SFNO | TFNO | UNet | DDPM |
|---|---|---|---|---|---|---|
| Tension | | | | | | |
| TS = 1 | MSE | $0.0019 \pm 0.0002$ | $0.0033 \pm 0.0001$ | $0.0142 \pm 0.0002$ | $0.0418 \pm 0.0001$ | $0.0008 \pm 0.0020$ |
| | L2 | $0.2653 \pm 0.0001$ | $0.1413 \pm 0.0001$ | $0.2613 \pm 0.0002$ | $0.3102 \pm 0.0300$ | $0.0760 \pm 0.0070$ |
| TS = 3 | MSE | $0.0141 \pm 0.0002$ | $0.0030 \pm 0.0002$ | $0.0039 \pm 0.0002$ | $0.0501 \pm 0.0300$ | $0.0018 \pm 0.0030$ |
| | L2 | $0.2938 \pm 0.0001$ | $0.1312 \pm 0.0001$ | $0.1530 \pm 0.0002$ | $0.3312 \pm 0.0030$ | $0.1114 \pm 0.0050$ |
| TS = 5 | MSE | $0.0084 \pm 0.0002$ | $0.0026 \pm 0.0002$ | $0.0032 \pm 0.0001$ | $0.0515 \pm 0.0003$ | $0.0027 \pm 0.0008$ |
| | L2 | $0.2356 \pm 0.0003$ | $0.1192 \pm 0.0001$ | $0.1335 \pm 0.0005$ | $0.3516 \pm 0.0200$ | $0.1375 \pm 0.0200$ |
| TS = 7 | MSE | $0.0078 \pm 0.0001$ | $0.0019 \pm 0.0002$ | $0.0024 \pm 0.0001$ | $0.0549 \pm 0.0002$ | $0.0034 \pm 0.0007$ |
| | L2 | $0.2243 \pm 0.0002$ | $0.0936 \pm 0.0002$ | $0.1038 \pm 0.0002$ | $0.3619 \pm 0.0010$ | $0.1557 \pm 0.0200$ |
| TS = 9 | MSE | $0.0049 \pm 0.0002$ | $0.0020 \pm 0.0001$ | $0.0031 \pm 0.0001$ | $0.0575 \pm 0.0003$ | $0.0043 \pm 0.0007$ |
| | L2 | $0.1915 \pm 0.0002$ | $0.1006 \pm 0.0002$ | $0.1167 \pm 0.0001$ | $0.3412 \pm 0.0001$ | $0.1708 \pm 0.0400$ |
| Wet | | | | | | |
| TS = 1 | MSE | $0.0784 \pm 0.0002$ | $0.0795 \pm 0.0001$ | $0.0957 \pm 0.0001$ | $0.0392 \pm 0.0004$ | $0.0171 \pm 0.0060$ |
| | L2 | $0.3720 \pm 0.0001$ | $0.4015 \pm 0.0002$ | $0.4383 \pm 0.0002$ | $0.5047 \pm 0.0040$ | $0.3051 \pm 0.0400$ |
| TS = 3 | MSE | $0.0704 \pm 0.0002$ | $0.0743 \pm 0.0002$ | $0.0817 \pm 0.0001$ | $0.0411 \pm 0.0002$ | $0.0227 \pm 0.0050$ |
| | L2 | $0.3052 \pm 0.0001$ | $0.3127 \pm 0.0001$ | $0.3417 \pm 0.0002$ | $0.5232 \pm 0.0030$ | $0.3234 \pm 0.0400$ |
| TS = 5 | MSE | $0.0871 \pm 0.0002$ | $0.0819 \pm 0.0002$ | $0.0883 \pm 0.0001$ | $0.0425 \pm 0.0020$ | $0.0228 \pm 0.0030$ |
| | L2 | $0.3887 \pm 0.0002$ | $0.3936 \pm 0.0001$ | $0.3585 \pm 0.0001$ | $0.5432 \pm 0.0030$ | $0.3418 \pm 0.0300$ |
| TS = 7 | MSE | $0.0846 \pm 0.0002$ | $0.0867 \pm 0.0002$ | $0.0945 \pm 0.0001$ | $0.0431 \pm 0.0003$ | $0.0231 \pm 0.0020$ |
| | L2 | $0.4039 \pm 0.0001$ | $0.3956 \pm 0.0001$ | $0.3788 \pm 0.0001$ | $0.5532 \pm 0.0020$ | $0.3516 \pm 0.0300$ |
| TS = 9 | MSE | $0.0860 \pm 0.0002$ | $0.0916 \pm 0.0002$ | $0.1072 \pm 0.0001$ | $0.0437 \pm 0.0030$ | $0.0232 \pm 0.0030$ |
| | L2 | $0.4034 \pm 0.0001$ | $0.4014 \pm 0.0001$ | $0.4141 \pm 0.0001$ | $0.6012 \pm 0.0020$ | $0.3777 \pm 0.0200$ |

Table 3: Comparison of benchmarks in terms of MSE and Relative L2 norm on Jellyfish (Fluid).

| Time Step | Metrics | FNO | SFNO | TFNO | UNet | DDPM |
|---|---|---|---|---|---|---|
| TS = 1 | MSE | $0.0319 \pm 0.0002$ | $0.0241 \pm 0.0003$ | $0.0228 \pm 0.0002$ | $0.6342 \pm 0.0002$ | $0.6970 \pm 0.0020$ |
| | L2 | $0.3127 \pm 0.0002$ | $0.2733 \pm 0.0001$ | $0.2646 \pm 0.0001$ | $0.9487 \pm 0.0002$ | $0.9520 \pm 0.0030$ |
| TS = 3 | MSE | $0.0524 \pm 0.0001$ | $0.0418 \pm 0.0003$ | $0.0421 \pm 0.0001$ | $0.6364 \pm 0.0003$ | $0.7021 \pm 0.0030$ |
| | L2 | $0.3991 \pm 0.0002$ | $0.3579 \pm 0.0001$ | $0.3549 \pm 0.0001$ | $0.9501 \pm 0.0001$ | $0.9601 \pm 0.0020$ |
| TS = 5 | MSE | $0.0793 \pm 0.0002$ | $0.0692 \pm 0.0001$ | $0.0718 \pm 0.0002$ | $0.6395 \pm 0.0002$ | $0.7073 \pm 0.0010$ |
| | L2 | $0.4859 \pm 0.0001$ | $0.4548 \pm 0.0001$ | $0.4496 \pm 0.0001$ | $0.9557 \pm 0.0001$ | $0.9612 \pm 0.0010$ |
| TS = 7 | MSE | $0.1004 \pm 0.0002$ | $0.0911 \pm 0.0003$ | $0.0997 \pm 0.0002$ | $0.6460 \pm 0.0001$ | $0.7145 \pm 0.0020$ |
| | L2 | $0.5416 \pm 0.0002$ | $0.5277 \pm 0.0001$ | $0.5435 \pm 0.0001$ | $0.9536 \pm 0.0001$ | $0.9634 \pm 0.0010$ |
| TS = 9 | MSE | $0.1253 \pm 0.0001$ | $0.1273 \pm 0.0001$ | $0.1164 \pm 0.0002$ | $0.6660 \pm 0.0002$ | $0.7167 \pm 0.0020$ |
| | L2 | $0.6044 \pm 0.0002$ | $0.6142 \pm 0.0001$ | $0.5918 \pm 0.0001$ | $0.9574 \pm 0.0002$ | $0.9679 \pm 0.0002$ |

perceptrons (MLPs). MLP-Mixer comprises two distinct layer types: one applies MLPs individually to image patches, while the other applies MLPs across multiple patches. DDOs-FNO (Lim et al., 2023) proposes a robust mathematical framework specifically designed for training diffusion models in function space. DDOs-SFNO (Lim et al., 2023) is inspired by the combination of DDPM and FNO, it is combination of DDPM and SFNO. DDOs-TFNO (Lim et al., 2023) is a hybrid approach that draws inspiration from both DDPM and TFNO. Detailed illustrations of benchmark methods and setups are shown in Appendix A.4.

**Results: Overall Comparison of Benchmark Methods.** Tables 2 and 3 present a comparative analysis of single-cell trajectory simulation performance across three distinct datasets: Tension, Wets, and Jellyfish. Three distinct classes of models were evaluated: autoregressive models (FNO, SFNO, TFNO, and UNet) and a generative model (DDPM). The results, quantified using Mean Squared Error (MSE) and Relative L2 norm, reveal a complex interplay between model architecture and dataset characteristics. While the autoregressive models demonstrate varying degrees of success across the datasets, a clear, consistent superiority over the generative model (DDPM) is not evident. Interestingly, a closer examination suggests that autoregressive models frequently exhibit superior simulation performance as measured by both MSE and Relative L2 norm.

**Visualization of DDPM.** Within our research, we offer visualizations using DDPM to present the ground truth, simulation results, and error analysis in Figure 2. The initial subfigure in the first column portrays the actual cell scenario, serving as the reference or truth. Moving to the second column, the subfigures showcase the cells generated through the employment of DDPM. Finally, in the third column, we observe the error figures, which highlight the disparities between the ground truth and the simulated results obtained using DDPM. Due to page limits, additional experimental results for six benchmark methods (*ConvLSTM*, *ViT*, *MLP-Mixer*, *DDOs-FNO*, *DDOs-SFNO* and *DDOs-TFNO*) can be found in Appendix A.5.

Figure 3 presents visualizations generated using DDPM applied to the CellDivision dataset. These visualizations demonstrate that DDPM effectively captures the complex dynamics of cell division within this dataset, generating simulated cell division patterns that closely resemble the observed

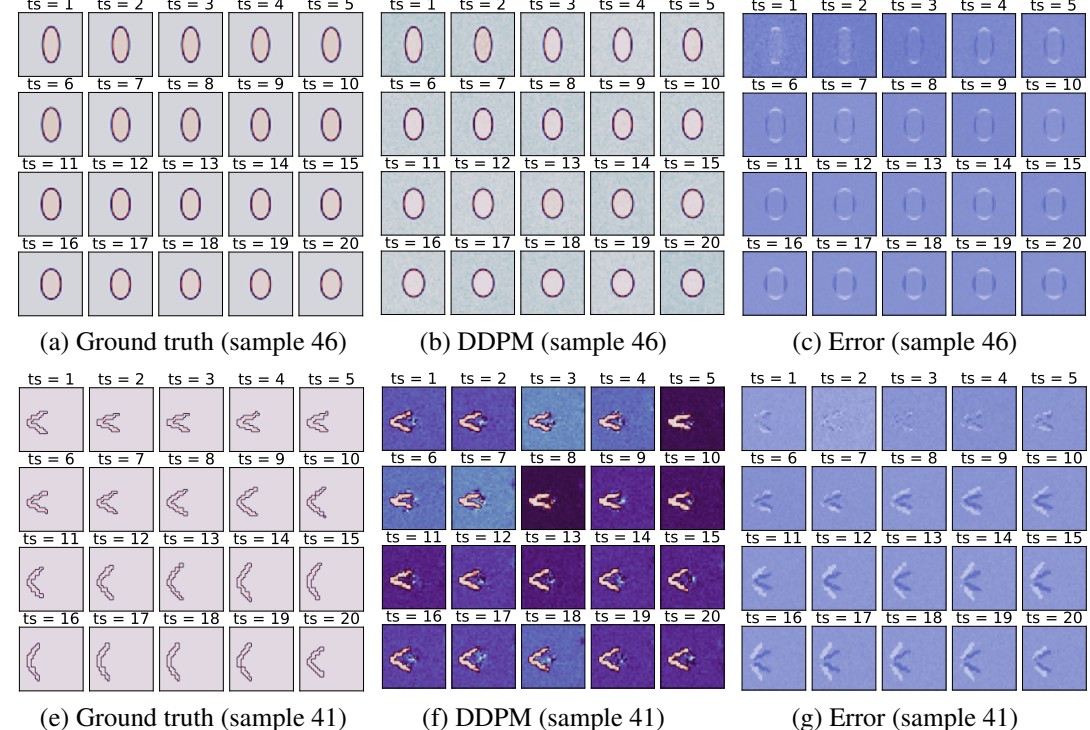

(a) Ground truth (sample 46)  (b) DDPM (sample 46)  (c) Error (sample 46)

(e) Ground truth (sample 41)  (f) DDPM (sample 41)  (g) Error (sample 41)

Figure 2: In our research, we provide visualizations using DDPM to examine the ground truth, simulation results, and error analysis on Tension and Jellyfish (Fluid) dataset. The first column's initial subfigure represents the ground truth like genuine boundary of a jelly-like robot. Moving to the second column, the subfigures display the simulated results via DDPM like the pressure fields. Finally, in the third column, we observe the error figures that emphasize the differences between the ground truth and the simulated results obtained with DDPM.

biological processes. The high fidelity of the DDPM-generated simulations suggests its potential as a valuable tool for modeling and understanding cell division mechanisms.

## 5 RELATED WORK

**Comparison with Existing Biological Dynamics Datasets**. With the rapid advancement of AI in scientific research, comprehensive biological datasets are becoming increasingly important, as evidenced by recent works such as (Dannenfelser et al., 2024; Pan et al., 2003; Gilpin et al., 2020; Kucera et al., 2024). However, current biological datasets face several limitations and are unable to fully meet the rapidly evolving demands of AI in the biological domain. For instance, Dannenfelser *et al.* (Dannenfelser et al., 2024) have introduced FlaMBé (Flow annotations for Multiverse Biological entities), a comprehensive compilation of meticulously curated datasets that encompass a diverse range of tasks aimed at capturing procedural knowledge found in biomedical texts. This dataset is driven by the realization that academic papers, particularly the methodologies sections, serve as a significant but disorganized source of procedural knowledge. However, FlaMBé is limited by its small scale and lack of cellular-level data, and it fails to capture diverse cell lineages and fates under varying developmental conditions. Additionally, Kucera *et al.* (Kucera et al., 2024) have proposed a Python software package that simplifies the creation of datasets and the evaluation of models for deep learning on protein structures. But it does not cover biological dynamics in terms of cells.

Motivated by the limitations above, our work aims to provide benchmark datasets that Tension, Wets and CellDivision , which focus on cell evolution in terms of cell lineages, cell fates and developing conditions. In addition, Jellyfish dataset covers the jellyfish moving process in terms of water pressure. These datasets incorporate the evolving trends of time in both the jellyfish robot's angle and the time-evolving data-driven biological simulation. Thus, our contribution not only provides three large-scale biological datasets but also establishes benchmarks for simulating these datasets within the realm of cell evolution.

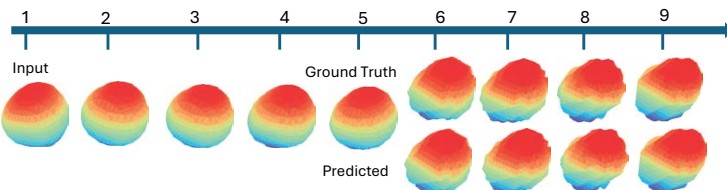

Figure 3: In our research, we provide visualizations using DDPM to examine the ground truth, simulation results for the CellDivision Dataset. The model uses a five-time-step cell evolution process as input to predict the subsequent four time steps of a cell's (Cell ABpr) lifecycle.

## 6   POTENTIAL APPLICATIONS

**Blood Flow in Cardiovascular Studies**: The Tension and Wets dataset, and our datasets more broadly, offer a valuable in silico model for investigating the intricate hemodynamics of blood flow within the cardiovascular system. The dataset's high fidelity allows for the simulation of complex fluid dynamics, including the effects of non-Newtonian fluid properties and vessel geometry. This capability enables the development and validation of AI-driven predictive models capable of assessing the impact of various pathophysiological factors, such as atherosclerotic plaque formation, valvular dysfunction, or stenosis, on blood flow patterns and shear stress distributions. Such models have the potential to significantly advance the design and optimization of medical interventions, including the planning of surgical procedures or the development of novel therapeutic strategies. Furthermore, the simulated data can be used to assess the efficacy of existing and novel treatments by providing a controlled environment for evaluating their impact on hemodynamic parameters. This approach offers a powerful complement to in vivo and in vitro studies, potentially reducing the reliance on animal models and accelerating the translation of research findings into clinical applications.

**The Motion of Swimming Organisms**: The Jellyfish dataset offers a rich resource for studying the complex fluid-structure interactions driving jellyfish locomotion. Analyzing this dataset reveals details of their propulsive mechanisms, including morphology, contractions, and fluid dynamics. This understanding informs bio-inspired robotics, improving underwater vehicle design. Furthermore, it enhances ecological modeling and our knowledge of marine life biophysics, impacting biomimetic material development for improved hydrodynamic properties.

**Cell Tracking**: The CellDivision dataset holds significant potential across diverse biological and computational research areas. Its high-resolution, biologically realistic simulations facilitate the development and validation of novel cell division models, allowing researchers to test hypotheses about the underlying mechanisms. The dataset can be used to train and evaluate machine learning algorithms for cell segmentation, tracking, and fate prediction, improving the accuracy and efficiency of image analysis in experimental studies. Furthermore, it enables the investigation of the effects of genetic mutations or environmental changes on cell division dynamics, providing insights into disease development and therapeutic interventions. The dataset's potential extends to the design of virtual experiments, reducing the need for costly and time-consuming laboratory work while allowing for controlled exploration of a wider range of conditions. Finally, it can serve as a benchmark for evaluating the performance of different computational methods in cell biology.

## 7   CONCLUSION AND LIMITATIONS

This paper addresses the challenges faced by the community in biological fluid simulation. Existing datasets lack the ability to capture dynamic processes and are limited in scale. To overcome these limitations, we introduce four comprehensive datasets: Tension, Wets, CellDivision and Jellyfish. These datasets push the boundaries of data-driven methods by encompassing a wide array of biological fluid dynamics scenarios. We also provide a standardized evaluation framework for data-driven approaches. The availability of these benchmark datasets enables objective assessment and comparison of methodologies, promoting advancements in the field. Additionally, the datasets enhance reproducibility and comparability of results, fostering collaboration and knowledge sharing. While these datasets have limitations in representing all phenomena and scalability, they serve as valuable resources for researchers to develop accurate and efficient models for simulating complex fluid dynamics in biological systems. Detailed limitations and broader impacts are shown in Appendix A.6.

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

# A APPENDIX

## A.1 RELATED WORK

**Diffusion Probabilistic Model**. Diffusion Probabilistic Models (DM) Ho et al. (2020) have emerged as the leading approach in density estimation Kingma et al. (2021) and have also demonstrated superior sample quality Dhariwal & Nichol (2021). These models leverage the inherent characteristics of image-like data by employing a UNet as their underlying neural backbone Ronneberger et al. (2015); Ho et al. (2020); Dhariwal & Nichol (2021). Notably, the use of a reweighted objective Ho et al. (2020) during training typically leads to the highest synthesis quality. Another research line for image generation is GAN-based methods Creswell et al. (2018); Chen et al. (2016b). A representative study of this research line is infoGAN Chen et al. (2016b), which is a type of generative adversarial network that not only generates realistic samples but also maximizes the mutual information between a select few latent variables and the generated output. InfoGAN allows for the discovery of meaningful representations. However, these studies cannot provide explanations for the generated samples.

**Denosing Diffusion Probabilistic Model (DDPM) for AI for Science**. Denoising diffusion probabilistic models have demonstrated their ability to predict dynamic evolution in a wide range of domains, including fluid dynamics Cachay et al. (2023), weather forecasting Price et al. (2023), and molecular dynamics Wu et al. (2022). They have also proven effective in inverse design tasks, facilitating the optimization of airfoils Wu et al. (2024) and proteins Watson et al. (2023). Additionally, diffusion models have shown promise in tackling complex inverse problems Holzschuh et al. (2023). These examples represent just a fraction of the diverse applications where diffusion models have been successfully employed. In the field of biology, researchers have utilized denoising diffusion probabilistic models (DDPM) to model diffusion processes in biological networks, enabling the analysis of protein-protein interactions and gene regulatory networks Fu et al. (2023); Best & Hummer (2011); Gao et al. (2023); Xu et al. (2022). In physics, the DDPM has been applied to study particle diffusion in complex systems, such as the propagation of heat in materials. Furthermore, in the realm of chemistry, the DDPM has been employed to gain insights into the diffusion of molecules and reactions in chemical systems. These studies highlight the versatility and effectiveness of the DDPM in capturing and analyzing diffusion dynamics across various scientific disciplines Xu et al. (2022). Ongoing research aims to further explore the potential of DDPMs in solving complex problems in the field of AI for Science. For additional related work on diffusion models, please refer to Appendix A.1.

## A.2 PRELIMINARY

**Diffusion Probabilistic Models:** The Denoising Diffusion Probabilistic Model (DDPM) Ho et al. (2020) comprises two fundamental processes: the forward process (or diffusion process) and the reverse process. Let's begin by describing the forward process. In a diffusion model, the forward process approximates the posterior distribution $q(x_{1:t}|x_0)$, which represents the sequence of latent variables $x_{1:t}$ given an initial value $x_0$. This approximation is achieved by iteratively applying a Markov chain that gradually adds Gaussian noise over time.

The forward process is represented as follows:

$$q(x_t|x_{t-1}) = \mathcal{N}(x_t; \mu_t(x_{t-1}), \beta_t I)$$

Here, $x_t$ denotes the latent variable at time step $t$, and $x_{t-1}$ is the variable at the previous time step. The distribution $q(x_t|x_{t-1})$ is modeled as a Gaussian distribution with mean $\mu_t(x_{t-1})$ and variance $\beta_t I$, where $\beta_t$ is the variance parameter at time step $t$, and $I$ represents the identity matrix. The mean $\mu_t(x_{t-1})$ can depend on the previous latent variable $x_{t-1}$ and is typically modeled using neural networks or other parameterized functions.

By sequentially applying the distribution $q(x_t|x_{t-1})$ for each time step, starting from the initial value $x_0$, we obtain an approximation of the posterior distribution $q(x_{1:t}|x_0)$ that captures the temporal evolution of the latent variables via $q(x_{1:t}|x_0) := \prod_{t=1}^{T} q(x_t|x_{t-1})$.

Consider a diffusion model with $T$ time steps. Given an observed data point $x_t$ at the final time step, the goal is to generate a sample from the initial distribution $p(x_0)$.

The reverse process in a diffusion model can be formulated as follows: 1. Initialization: Set $x_t$ as the observed data point. 2. Iterative Sampling: Starting from $t = T - 1$ and moving backwards until

$t = 0$, sample $x_t$ from the distribution $p(x_t|x_{t+1})$, where $p(x_t|x_{t+1})$ represents the reverse diffusion process.

The distribution $p(x_t|x_{t+1})$ in the reverse process is typically modeled as a Gaussian distribution, similar to the forward process. However, the mean and variance parameters are adjusted to account for the reverse direction. The specific form of $p(x_t|x_{t+1})$ is defined as follows:

$$p_{t+1}(x_t; \mu_{t+1}(x_{t+1}, t+1), \Sigma_{t+1}(x_{t+1}, t+1))$$

By iteratively sampling from the reverse process, we can generate a sequence of latent variables $x_{0:t}$ that follows the reverse diffusion process. This reverse sequence represents a sample from the initial distribution $p(x_0)$. The reverse process is crucial for training the diffusion model. During training, the model learns to approximate the reverse process by minimizing the discrepancy between the generated samples and the observed data points. This training procedure ensures that the model captures the underlying data distribution and can generate realistic samples.

The optimization objective of the diffusion model is conducted via the following negative log-likelihood:

$$E[-\log p_{t+1}(x_0)] \leq E_{p_{t+1}}[-\log(p_{t+1}(x_{0:t})/q(x_{1:t}|x_0))]$$

$$= E_{p_{t+1}}[-\log p_{t+1}(x_t) - \sum_{t=1}^{T} \log(p_{t+1}(x_{t-1}|x_t)/q(x_t|x_{t-1}))] = L$$

**Cahn-Hilliard Function:** Wetting phenomena and interfacial tension play significant roles in numerous scientific and engineering fields, ranging from fluid dynamics to materials science. In recent years, phase-field methods have emerged as powerful computational tools for studying and simulating wetting processes. These methods employ a phase-field variable, a continuous function that describes the local composition or wetting state, enabling the realistic modeling of complex interfacial dynamics. By incorporating the concept of interfacial tension, phase-field models can capture the intricate interplay between fluids and solid surfaces.One of the key equations used in phase-field modeling of wetting phenomena is the Cahn-Hilliard equation, which governs the evolution of the phase-field variable. This equation takes into account the interfacial energy associated with the fluid-solid interface and the interfacial tension between the two phases. The interfacial tension term is crucial for accurately simulating the contact angle, adhesion, and spreading behavior. The Cahn-Hilliard equation provides a mathematical framework to capture the dynamics of phase separation by considering the free energy of the system. It takes the following general form:

$$\frac{\partial \phi}{\partial t} = \nabla \cdot \left( M \nabla \left( \frac{\delta F}{\delta \phi} \right) \right) \tag{5}$$

where $\phi$ is the phase-field variable or order parameter representing the local composition. $t$ is time. $M$ is the mobility coefficient, controlling the rate of diffusion of the phase-field variable. $F$ is the free energy functional of the system with respect to the phase-field variable $\phi$. The free energy functional $F$ typically consists of two terms: the bulk free energy term and the gradient energy term. The bulk free energy accounts for the thermodynamic properties of the system, including the interfacial energy between the two phases and the potential energy associated with phase separation. The gradient energy term penalizes sharp variations or spatial gradients in the order parameter, promoting smoother phase transitions.

**Functional Formulations for Modeling Tension Phenomena via Phase-Field:** To accurately represent the shape of an arbitrary object, we utilize a phase-field variable $\phi$, where $\phi = 1$ denotes the interior of the cell and $\phi = 0$ denotes the exterior. The transition from $\phi = 1$ to $\phi = 0$ occurs gradually within a width defined by the parameter $\epsilon$. The system's total energy is denoted by $H(\phi)$, and the time evolution of $\phi$ is determined by the following equation $\frac{\partial \phi}{\partial t} = -\frac{\delta H(\phi)}{\delta \phi}$. This equation describes the rate of change of $\phi$ with respect to time. The right-hand side represents the derivative of the total energy $H(\phi)$ with respect to $\phi$, indicating the force or driving mechanism that governs the evolution of the phase-field variable $\phi$. By minimizing the energy functional $H(\phi)$, the system tends to reach an equilibrium state that corresponds to the desired shape of the object. For the tension dataset, the surface tension is characterized by the tension per unit length multiplied by the total surface area. To ensure proper normalization, we express it as follows:

$$H_{\text{ten}}(\phi) = \gamma \int d^2 r \left( \frac{\epsilon}{2} |\nabla \phi|^2 + \frac{G(\phi)}{\epsilon} \right) \tag{6}$$

Here, $\gamma$ denotes the coefficient of surface tension, while $\epsilon$ represents the characteristic width of the interface. The term $|\nabla\phi|^2$ quantifies the gradient of $\phi$ in space, while $G(\phi)$ is a function proportional to the perimeter of the object and is given by $G(\phi) = 18\phi^2(1 - \phi)^2$. The integral in Equation 6 accounts for the total energy associated with surface tension. The time evolution under tension is described by the following equation, known as the Allen-Cahn equation, which is a reaction-diffusion equation:

$$\frac{\partial\phi}{\partial t} = -\frac{\delta H_{\text{ten}}(\phi)}{\delta t} = -\gamma\left(\epsilon\nabla^2\phi + \frac{G'(\phi)}{\epsilon}\right) \quad (7)$$

In practice, the term $G(\phi)$ yields similar results to the $|\nabla\phi|^2$ term. Therefore, when explicitly calculating the total tension, we can use the following simplified form:

$$H'_{\text{ten}}(\phi) = \gamma \int d^2r \left(\frac{2G(\phi)}{\epsilon}\right) \quad (8)$$

Equation 8 provides an alternative expression for the total tension, which considers only the $G(\phi)$ term. This formulation allows for efficient computation of the tension without explicitly calculating the gradient term.

**Functional Formulations of Wets via Phase-field:** The interaction of an object with a substrate involves adhesion, which pulls the object towards the substrate, and a repellent force that prevents the object from penetrating into the substrate. To express the total energy in a physically consistent manner, we define it as follows:

$$H_{\text{sub}}(\phi) = \gamma \int d^2r \left(-A\frac{2G(\phi)G(\varphi)}{\epsilon^2} + B\phi\varphi\right) \quad (9)$$

Here, $\gamma$ represents the coefficient of adhesion, $A$ is the adhesion strength (with $A > 0$), and $B$ is the repellent strength (with $B > 0$). The field $\varphi$ corresponds to the substrate. In our simulation, the substrate is considered a fixed function given by:

$$\varphi(r) = \frac{1}{2}\left\{1 + \tanh\left[3 \times \left(\frac{y_0 - y}{\epsilon}\right)\right]\right\} \quad (10)$$

Here, the substrate is positioned at a specific vertical location, $y = y_0$. In our simulation, we set $y_0 = -10$. The function $\varphi$ describes the spatial profile of the substrate, with a smooth transition from high to low values as $y$ increases from the substrate position. During this period, the evolution function, which includes the tension part, is given by:

$$\frac{\partial\phi}{\partial t} = -\gamma(\epsilon\nabla^2\phi + \frac{G'(\phi)}{\epsilon}) - A\frac{G'(\phi)G(\varphi)}{\epsilon^2} + B\varphi \quad (11)$$

Equation 11 represents the time derivative of $\phi$, where the first term on the right-hand side accounts for the tension contribution, the second term describes the adhesion interaction between the object and the substrate, and the third term represents the repellent force due to the substrate. This evolution equation governs the dynamics of the phase-field variable $\phi$ in the presence of adhesion and substrate effects.

A.3 DATASETS GENERATION

**Tension Datasets Generation:** (1) The configuration of the grid and domain: The parameters $m$ and $n$ determine the quantity of grid points along the $x$ and $y$ axes, respectively. The dimensions of the domain are specified by $L_x$ and $L_y$ in the $x$ and $y$ directions. Vectors $x$ and $y$ denote the coordinates of the grid points along the x and y axes, correspondingly. The parameters $k$ and $\epsilon$ regulate the bending and tension within the system. (2) $ten_{vec}$ is a vector that iterates over tension values for the simulation. Inside the loop for each tension value: $\gamma$ is the tension parameter. $dt$, $x_i$, $M_v$ are time steps, smoothing parameter, and viscosity parameters, respectively. $N_{max}$ is the maximum number of iterations. $record_{num}$ determines how often the simulation results are recorded. $x_{radi}, y_{radi}, r_{radi}$ are parameters defining the initial shape of the system. $\phi_0$ is the initial condition of the simulation. (3) Change $m, n, L_x, L_y$ for a finer or coarser grid or a larger/smaller domain. Modify $k, \epsilon$, and other parameters to explore different physical scenarios. Adjust parameters inside the tension loop ($dt, x_i, M_v$, etc) for different simulation characteristics. Change the initial shape parameters ($x_{radi}, y_{radi}, r_{radi}$) to explore different starting configurations.

**Wets Datasets Generation:** (1) The vector $ten_{vec}$ represents tension values used in the simulation, while $adh_{vec}$ represents adhesion values. For each combination of tension and adhesion values, denoted by $ten_i$ and $adh_j$ iterating through the indices of $ten_{vec}$ and $adh_{vec}$ respectively, the following actions take place. The parameter $\gamma$ is set to the current tension value, and $adh$ is set to the current adhesion value. The variable $rep$ is calculated as a multiple of tension. At the beginning of each iteration, the initial shape parameters $(x_{radi}, y_{radi}, r_{radi})$ and the shape initialization $(\phi_0)$ are redefined to ensure unique initial conditions for each combination of tension and adhesion values. The position of the substrate is represented by $y_0$, and the substrate's initial shape $sub_0$ is initialized using a hyperbolic tangent function. (2) By altering these parameters, particularly adjusting tension, adhesion, and other physical factors, you have the ability to generate datasets that depict wetting in various scenarios. Depending on your requirements, you can experiment with different levels of tension, adhesion, initial shapes, and simulation parameters to examine how the system's morphology evolves under different conditions.

**Jellyfish Datasets Generation:** To generate our training and testing datasets, we employ the Lily-Pad simulator Weymouth (2015). The 2D flow field has a resolution of $128 \times 128$, assuming an infinite extension. For the jellyfish, the fixed coordinates for the head are set as $(25.6, 64)$. The wings are represented as ellipses with an identical shape, with a fixed ratio of 0.15 between their shorter and longer axes. Symmetry is maintained across the central horizontal line defined by $y = 64$. To delineate the boundaries of the wings, we sample a total of $M = 20$ points along each wing. In this 2D experiment, the key control signal is the opening angle of the wings, denoted as $w$. This angle is defined as the deviation between the longer axis of the upper wing and the horizontal line.

Each trajectory commences with the widest possible opening angle and follows a periodic cosine curve with a period of $T' = 200$. The trajectories differ in their initial angle $(w_0)$, angle amplitude, and phase ratio $(\tau)$. To determine the initial angle $w_0$, a two-step process is employed. Firstly, a random mean angle $w^{(m)}$ is sampled from the range of $[20°, 40°]$. Then, a random angle amplitude $w^{(a)}$ is sampled from the interval $[10°, \min(w^{(m)}, 60° - w^{(m)})]$. The resulting initial angle is computed as $w_0 = w^{(m)} + w^{(a)}$, constrained within the range of $[10°, 60°]$. The phase ratio $\tau$ is randomly selected from the range of $[0.2, 0.8]$. The opening angle $w_t$ at step $t$ adheres to a specific pattern: it decreases from $w^{(m)} + w^{(a)}$ to $w^{(m)} - w^{(a)}$ as $t$ progresses from 0 to $\tau T'$, and then it increases from $w^{(m)} - w^{(a)}$ to $w^{(m)} + w^{(a)}$ as $t$ advances from $\tau T'$ to $T'$. Beyond $T'$, $w_t$ exhibits periodic variations. This configuration aligns with previous studies on jellyfish's propulsive performance Kang et al. (2023). Each trajectory is simulated for 600 steps, equivalent to 3 periods. Only the segment of the trajectory from $T' = 200$ to $3T' = 600$ steps is saved, with a step size of 10. This decision is made to conserve space, as the simulation from $t = 0$ to $T' = 200$ is primarily used for initializing the flow field. Consequently, each trajectory is stored as a sequence consisting of $\tilde{T} = (600 - 200)/10 = 40$ discrete steps.

In addition to tracking the positions of the wing boundary points and the opening angles $w$, we incorporate an image-like representation of the wing boundaries. This representation contains spatial information that can be efficiently integrated with the PDE states (fluid field) through convolutional neural networks. The image-like boundary representation seamlessly aligns with the shape of the PDE states. At each time step, the boundaries of the two wings are combined and transformed into a tensor with dimensions [3, 64, 64]. This tensor represents the spatial information in a grid-like format. Each cell in the tensor contains three features: a binary mask indicating whether the cell is part of a wing boundary (1) or within the fluid (0), and the relative position $(\Delta x, \Delta y)$, which denotes the distance from the cell center to the nearest boundary point. For each trajectory, the following components are saved: - PDE states $u$: This captures the fluid field states for each time step and has a shape of $[\tilde{T}, 3, 64, 64]$. It includes the velocity components in the $x$ and $y$ directions as well as the pressure. The resolution is downsampled from $128 \times 128$ to $64 \times 64$. - Velocity: $[\tilde{T}, 2, 64, 64]$. - Pressure: $[\tilde{T}, 1, 64, 64]$. - Opening angles $w$: This stores the opening angle in radians for each step and has a shape of $[\tilde{T}]$. - Boundary points: This records the boundary points for both the upper and lower wings and has a shape of $[\tilde{T}, 2, M, 2]$. Each wing consists of $M = 20$ points, and each point is represented by its coordinates in the $x$ and $y$ directions. The coordinates are scaled accordingly to fit within the grid dimensions.

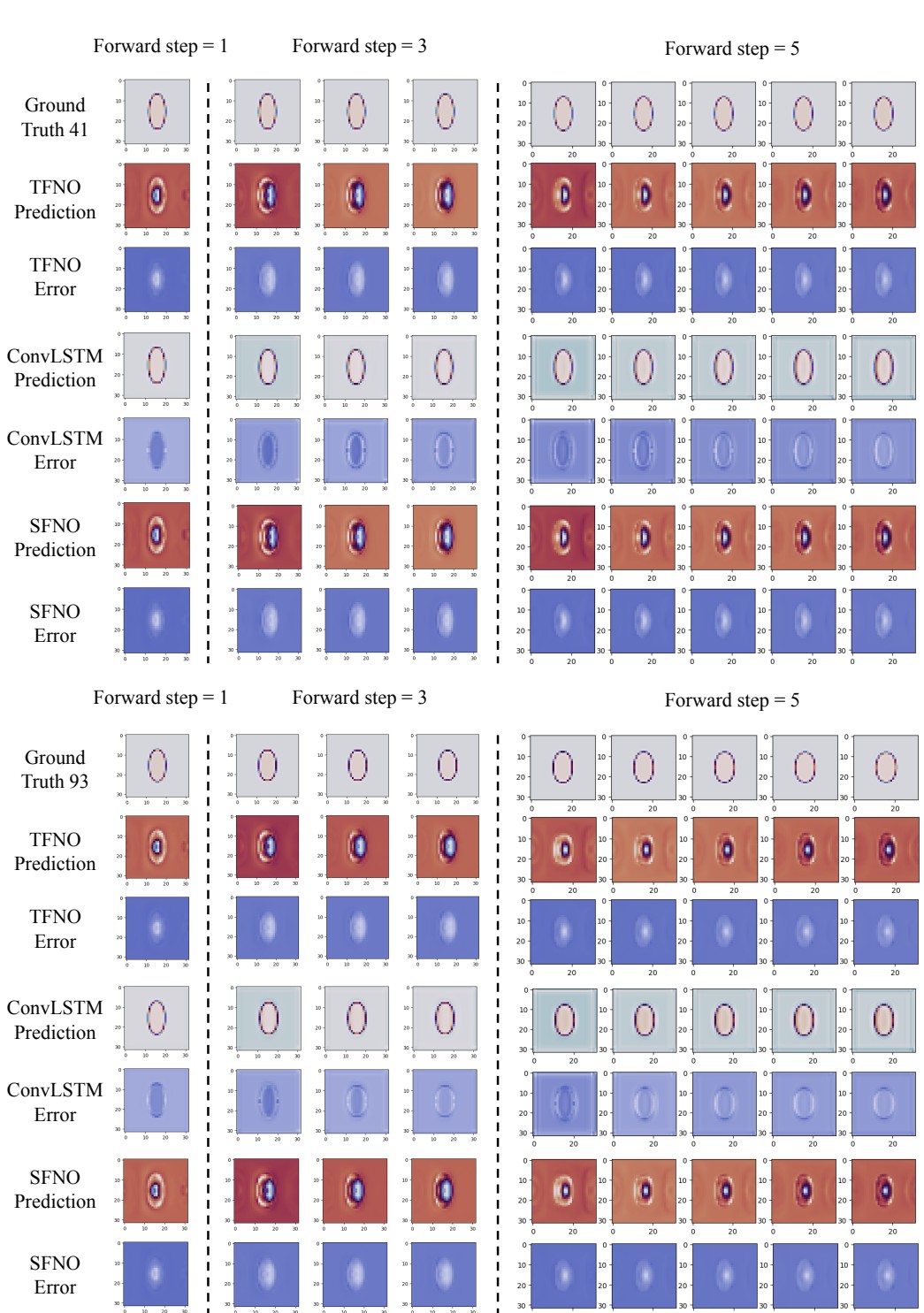

Figure 4: In our study, we offer visualizations that compare benchmark methods, namely TFNO, SFNO and ConvLSTM, using the tension dataset. The first row portrays the authentic cell scenario, which serves as the ground truth. The subsequent rows illustrate the cells generated by the benchmark methods and the corresponding error bar.

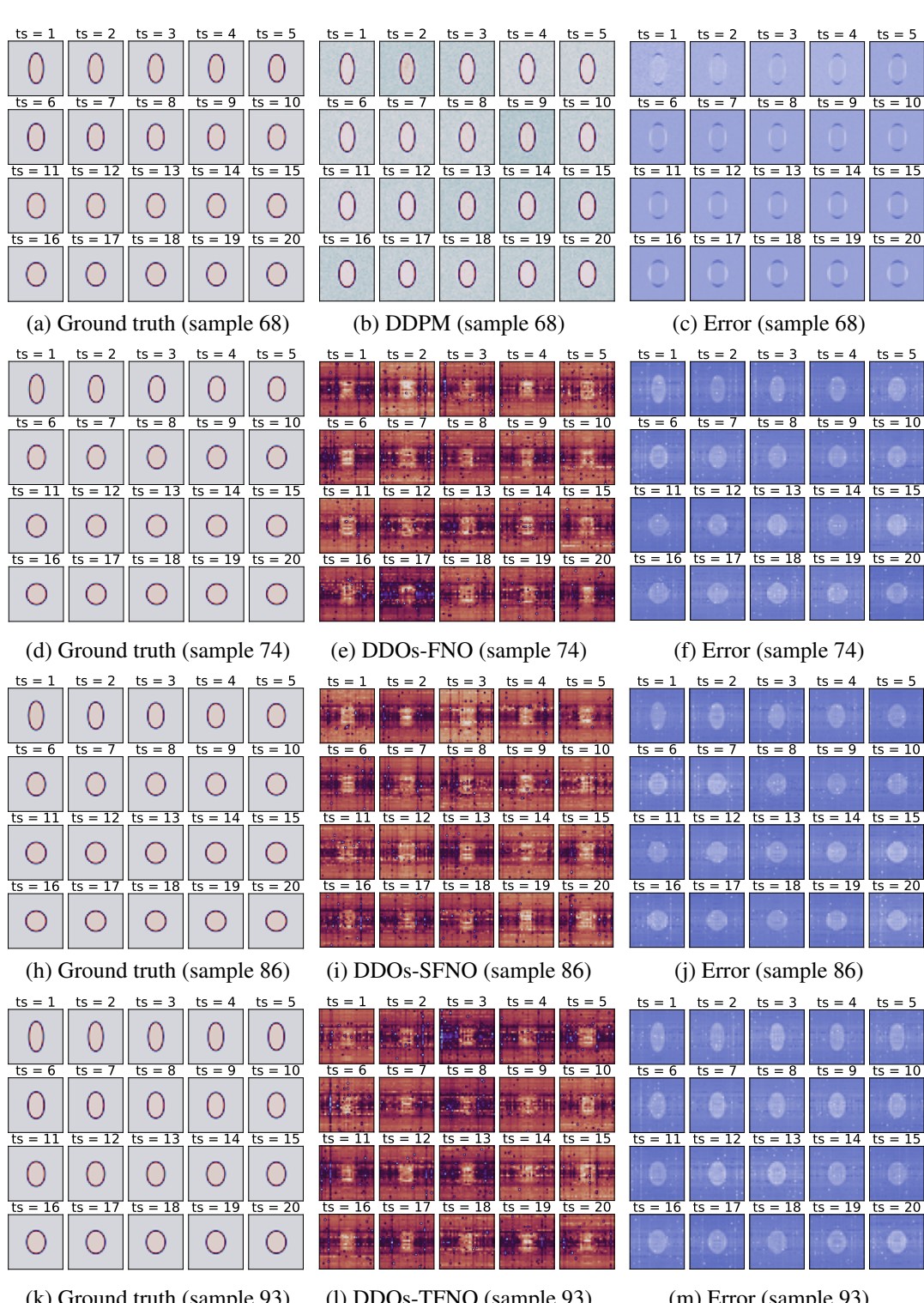

(a) Ground truth (sample 68) (b) DDPM (sample 68) (c) Error (sample 68)

(d) Ground truth (sample 74) (e) DDOs-FNO (sample 74) (f) Error (sample 74)

(h) Ground truth (sample 86) (i) DDOs-SFNO (sample 86) (j) Error (sample 86)

(k) Ground truth (sample 93) (l) DDOs-TFNO (sample 93) (m) Error (sample 93)

Figure 5: In our study, we present visualizations using DDPM, including the ground truth, simulation results, and error analysis. The first column subfigure represents the authentic cell scenario, which serves as the reference. The subfigures in the second column showcase the cells generated by DDPM. The subfigures in the third column illustrate the error figures, highlighting the differences between the ground truth and the simulation results obtained through DDPM.

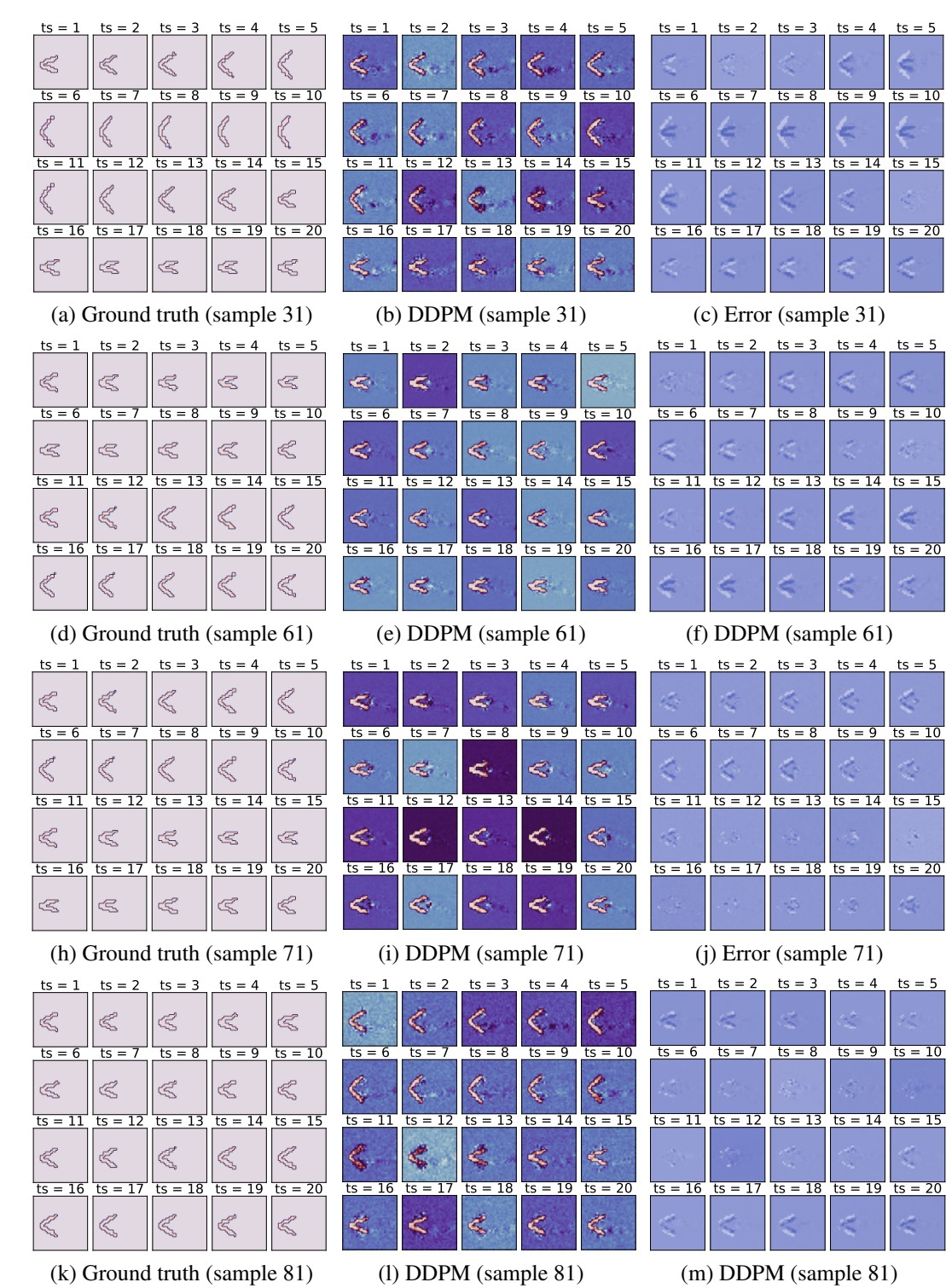

Figure 6: Our research includes visualizations utilizing DDPM to analyze the Jellyfish (Fluid) dataset, specifically focusing on the ground truth, simulation results, and error analysis. In the first column, the initial subfigure presents the accurate depiction of the boundary for a jelly-like robot. Transitioning to the second column, the subfigures demonstrate the pressure fields generated through the implementation of DDPM. Lastly, in the third column, the error figures provide a visual representation of the disparities between the ground truth and the simulated results obtained using DDPM.

## A.4 Detailed Illustration and Setup of Benchmark Methods

**Detailed Illustrations of Benchmark Methods:** In this part, we aim to provide detaield illustrations on baselines for simulating cell evolving.

**FNO Li et al. (2020):** The Fourier Neural Operator is a groundbreaking approach that utilizes Fourier space to learn weights, enabling resolution-invariant global convolutions. This influential work has been further expanded upon by numerous other neural operators. However, one notable drawback is the substantial memory requirements. Specifically, each weight matrix in the Fourier domain consumes $\mathcal{O}(H^2 M^D)$ memory, where $H$ represents the hidden size, $M$ denotes the number of Fourier modes used after truncating high frequencies, and $D$ signifies the problem dimension.

**SFNO Bonev et al. (2023):** Spherical Fourier Neural Operators utilizes spherical harmonics to transform data into the frequency domain. This is analogous to the way FNO uses the Fourier transform for Cartesian grids but is specifically designed for spherical data. This approach allows for handling data on the entire sphere without the distortions introduced by map projections. While the traditional FNO faces substantial memory demands, SFNO optimizes memory usage through spherical harmonics. Despite this, the memory requirement for SFNO is still $\mathcal{O}(H^2 M^2)$, where $H$ is the hidden size and $M$ is the number of retained spherical harmonic modes.

**TFNO:** Li et al. (2020) We improve the previous FNO model by simply using a Tucker Tensorized FNO with just a few parameters. This will use a Tucker factorization of the weights. The forward pass will be efficient by contracting directly the inputs with the factors of the decomposition. The Fourier layers will have 5% of the parameters of an equivalent, dense FNO.

**UNet Ronneberger et al. (2015):** The UNet architecture is known for its U-shaped design, which resembles an encoder-decoder structure. It is particularly effective for tasks that require precise localization and segmentation of objects within images. UNet has achieved remarkable performance in various applications, including biomedical image segmentation, satellite image analysis, and more. The Unet consists of the forward and backward passes. the time complexity is mainly driven by the size of the input image or volume, denoted as $\mathcal{O}(N)$, where $N$ is the number of pixels or voxels. It is important to note that the specific implementation details and variations of UNet may introduce additional computational complexities.

**DDPM:** Ho et al. (2020) It gradually transforms a noise-corrupted data version into the original distribution through iterative diffusion. Training involves two components: sampling and loss function computation. Sampling complexity is $\mathcal{O}(S * N)$, where $S$ is the diffusion steps and $N$ is the step size. The loss function compares generated samples with the original data, with complexity $\mathcal{O}(L * N)$, where $L$ is the number of layers. Overall training complexity is approximately $\mathcal{O}(S * N + L * N)$. DDPM models complex data distributions using denoising diffusion. Training time complexity is approximated as $\mathcal{O}(S * N + L * N)$, where S is diffusion steps, $L$ is layers, and $N$ is layer size.

**ConvLSTM** Shi et al. (2015) postulates the prognostication of forthcoming spatiotemporal precipitation patterns as an endeavor entailing the anticipation of sequential data in spatial and temporal dimensions.

**ViT** Dosovitskiy et al. (2020) asserts that the reliance on convolutional neural networks (CNNs) is dispensable, as the direct application of transformers to sequences of image patches yields exceptional performance in the classification of visual data.

**MLP-Mixer** Tolstikhin et al. (2021) is exclusively constructed upon the foundation of multi-layer perceptrons (MLPs). It encompasses two distinct layer types: one that individually applies MLPs to image patches, while the other employs MLPs across multiple patches, promoting enhanced representation learning.

**DDOs-FNO** Lim et al. (2023) advances a robust mathematical framework meticulously tailored for the training of diffusion models within the realm of function space.

**DDOs-SFNO** Lim et al. (2023) derives inspiration from the fusion of DDPM and FNO methodologies, forging a cohesive amalgamation of their respective strengths.

**DDOs-TFNO** Lim et al. (2023) adopts a hybrid approach, drawing inspiration from both DDPM and TFNO methods, thereby capitalizing on their synergistic potential.

**Setups of Benchmark Methods:**

Setting of **FNO**: The modes and width of FNO was set to 12 and 32, providing a suitable level of complexity for the specific task at hand. The number of channels for the two cell dynamics datasets and the fluid dataset was determined as $[1, 3, 5, 7, 9]$, taking into consideration the unique characteristics of each dataset. All three datasets were standardized to an image size of $32 \times 32$ pixels. In order to accelerate convergence, a learning rate of $1 \times 10^{-3}$ was adopted.

Setting of **SFNO**: The SFNO modes were set to 16, and the hidden channel was configured to 64. The channel numbers for both cell dynamics datasets and the fluid dataset were set to $[1, 3, 5, 7, 9]$, reflecting the unique characteristics of each dataset. All three datasets were standardized to an image size of $32 \times 32$ pixels. To accelerate convergence, we implemented a learning rate of $1 \times 10^{-3}$.

Setting of **TFNO**: The TFNO modes were set to 16, and the hidden channel was configured to 64. We employed Tucker factorization with a rank of 0.05. The channel numbers for both cell dynamics datasets and the fluid dataset were set to $[1, 3, 5, 7, 9]$, reflecting the unique characteristics of each dataset. All three datasets were standardized to an image size of $32 \times 32$ pixels. To accelerate convergence, we implemented a learning rate of $1 \times 10^{-3}$.

Setting of **UNet**: The hidden size of the UNet convolutional neural network was set to 32, providing a suitable level of complexity for the specific task at hand. The number of channels for the two cell dynamics datasets and the fluid dataset was determined as $[1, 3, 5, 7, 9]$, taking into consideration the unique characteristics of each dataset. All three datasets were standardized to an image size of $32 \times 32$ pixels. In order to accelerate convergence, a learning rate of $1 \times 10^{-4}$ was adopted.

Setting of **DDPM**: To ensure equity and uniformity in our experimental procedures, the concealed dimension of the U-Net convolutional neural network was established at 32, furnishing an apt level of intricacy for the given undertaking. For the Gaussian diffusion process, an extensive 1000 diffusion steps were executed, facilitating comprehensive exchange of information. The channel size multiplier of the U-Net neural networks was stipulated as $[1, 2, 4, 8]$, ensuring efficacious extraction of features across diverse scales. The number of channels for the two cell dynamics datasets and the fluid dataset were defined as 20, accommodating the distinctive attributes of each dataset. The dimensions of all three datasets were standardized to an image size of $32 \times 32$ pixels. To expedite convergence, a learning rate of $8 \times 10^{-5}$ was embraced.

Setting of **ConvLSTM**: The ConvLSTM model is initialized with input dimension and hidden dimension are both set to 1, indicating a single-channel input and a single-channel output per hidden layer. The model is designed to process sequences of length determined by $[1, 3, 5, 7, 9]$, representing the time dimension parameter. The convolution operation within the LSTM utilizes a kernel size of $3 \times 3$, which allows the model to capture spatial relationships within the data effectively.

Setting of **ViT**: ViT uses the temporal positional encoding method to handle the sequence length for positional encoding, applied across combined time steps and patches (with size equals to 4). A linear layer transforms the patch embeddings back into pixel values, ensuring the reconstruction of the original image or the generation of future frames in the biological trajectory. The channel numbers for both cell dynamics datasets and the fluid dataset were set to $[1, 3, 5, 7, 9]$, reflecting the unique characteristics of each dataset. All three datasets were standardized to an image size of $32 \times 32$ pixels. To accelerate convergence, we implemented a learning rate of $1 \times 10^{-3}$.

Setting of **MLP-Mixer**: The MLP-Mixer processes input data through two primary stages: channel-mixing and token-mixing, utilizing a patch size of $4 \times 4$, hidden dimensions of 32, and 4 layers. The number of channels for both cell dynamics datasets and the fluid dataset was set to $[1, 3, 5, 7, 9]$, tailored to the unique characteristics of each dataset. All three datasets were standardized to an image size of $32 \times 32$ pixels. To enhance convergence, a learning rate of $1 \times 10^{-3}$ was adopted.

Setting of **DDOs-FNO/DDOs-SFNO/DDOs-TFNO**: The hidden size of the FNO neural network in DDOs was set to 32, providing an appropriate level of complexity for the specific task at hand. To enable thorough information exchange, a substantial number of 1000 diffusion steps were performed for the Gaussian diffusion process. The number of channels for the two cell dynamics datasets and the fluid dataset was determined as 20, taking into account the unique characteristics of each dataset. All three datasets were standardized to an image size of $32 \times 32$ pixels. In order to accelerate convergence, a learning rate of $8 \times 10^{-5}$ was adopted.

## A.5 RESULTS OF BENCHMARK METHODS

We have presented the results of our experiments in Table 4 and Table 5. Upon careful examination of the tables, it becomes evident that ConvLSTM outperforms other models, primarily due to its inherent capability to capture long sequences effectively. Additionally, our observations reveal that DDOs-FNO, DDOs-SFNO, and DDOs-TFNO do not exhibit satisfactory performance across the three datasets. This can be attributed to the inherent challenges faced by these models in terms of convergence when applied to biological datasets.

Table 4: Comparison of benchmarks in terms of MSE and Relative L2 norm on two datasets.

| Time Step | Metrics | ConvLSTM | ViT | MLP-Mixer | DDOs-FNO | DDOs-SFNO | DDOs-TFNO |
|---|---|---|---|---|---|---|---|
| Tension | | | | | | | |
| TS = 1 | MSE | $0.0165 \pm 0.0003$ | $0.0869 \pm 0.0002$ | $0.0342 \pm 0.0001$ | $0.1288 \pm 0.0010$ | $0.1012 \pm 0.0020$ | $0.1622 \pm 0.0020$ |
| | L2 | $0.2982 \pm 0.0002$ | $0.7431 \pm 0.0001$ | $0.4065 \pm 0.0002$ | $0.7218 \pm 0.0020$ | $0.6022 \pm 0.0030$ | $0.6407 \pm 0.0010$ |
| TS = 3 | MSE | $0.0212 \pm 0.0002$ | $0.0842 \pm 0.0002$ | $0.0497 \pm 0.0002$ | $0.1295 \pm 0.0030$ | $0.1124 \pm 0.0010$ | $0.1734 \pm 0.0040$ |
| | L2 | $0.3564 \pm 0.0001$ | $0.7289 \pm 0.0001$ | $0.4760 \pm 0.0002$ | $0.7418 \pm 0.0030$ | $0.6228 \pm 0.0010$ | $0.6538 \pm 0.0010$ |
| TS = 5 | MSE | $0.0805 \pm 0.0001$ | $0.0839 \pm 0.0003$ | $0.0519 \pm 0.0001$ | $0.1306 \pm 0.0020$ | $0.1206 \pm 0.0020$ | $0.1801 \pm 0.0030$ |
| | L2 | $0.6086 \pm 0.0003$ | $0.7319 \pm 0.0001$ | $0.5369 \pm 0.0001$ | $0.7512 \pm 0.0030$ | $0.6322 \pm 0.0030$ | $0.6622 \pm 0.0010$ |
| TS = 7 | MSE | $0.0858 \pm 0.0002$ | $0.0849 \pm 0.0003$ | $0.1562 \pm 0.0003$ | $0.1311 \pm 0.0020$ | $0.1278 \pm 0.0030$ | $0.1846 \pm 0.0020$ |
| | L2 | $0.6455 \pm 0.0003$ | $0.8217 \pm 0.0002$ | $0.7913 \pm 0.0002$ | $0.7715 \pm 0.0010$ | $0.6401 \pm 0.0030$ | $0.6703 \pm 0.0030$ |
| TS = 9 | MSE | $0.0860 \pm 0.0002$ | $0.0884 \pm 0.0002$ | $0.1006 \pm 0.0002$ | $0.1387 \pm 0.0020$ | $0.1304 \pm 0.0020$ | $0.1887 \pm 0.0020$ |
| | L2 | $0.6688 \pm 0.0003$ | $0.7738 \pm 0.0002$ | $0.7754 \pm 0.0002$ | $0.7815 \pm 0.0020$ | $0.6517 \pm 0.0010$ | $0.6806 \pm 0.0010$ |
| Wet | | | | | | | |
| TS = 1 | MSE | $0.0798 \pm 0.0002$ | $0.1354 \pm 0.0001$ | $0.1282 \pm 0.0002$ | $0.1477 \pm 0.0030$ | $0.1319 \pm 0.0030$ | $0.1192 \pm 0.0030$ |
| | L2 | $0.3612 \pm 0.0001$ | $0.6897 \pm 0.0002$ | $0.6019 \pm 0.0002$ | $0.6209 \pm 0.0040$ | $0.6028 \pm 0.0030$ | $0.6011 \pm 0.0020$ |
| TS = 3 | MSE | $0.0843 \pm 0.0002$ | $0.1512 \pm 0.0002$ | $0.1267 \pm 0.0001$ | $0.1501 \pm 0.0020$ | $0.1489 \pm 0.0010$ | $0.1243 \pm 0.0020$ |
| | L2 | $0.3861 \pm 0.0001$ | $0.6958 \pm 0.0001$ | $0.5989 \pm 0.0002$ | $0.6235 \pm 0.0010$ | $0.6172 \pm 0.0010$ | $0.6224 \pm 0.0030$ |
| TS = 5 | MSE | $0.0899 \pm 0.0002$ | $0.1743 \pm 0.0002$ | $0.1325 \pm 0.0001$ | $0.1566 \pm 0.0020$ | $0.1512 \pm 0.0010$ | $0.1304 \pm 0.0010$ |
| | L2 | $0.3872 \pm 0.0001$ | $0.7341 \pm 0.0001$ | $0.6057 \pm 0.0001$ | $0.6308 \pm 0.0030$ | $0.6172 \pm 0.0010$ | $0.6406 \pm 0.0030$ |
| TS = 7 | MSE | $0.0965 \pm 0.0002$ | $0.1765 \pm 0.0002$ | $0.1369 \pm 0.0001$ | $0.1602 \pm 0.0020$ | $0.1599 \pm 0.0030$ | $0.1368 \pm 0.0020$ |
| | L2 | $0.3946 \pm 0.0002$ | $0.7355 \pm 0.0001$ | $0.6129 \pm 0.0001$ | $0.6405 \pm 0.0020$ | $0.6509 \pm 0.0030$ | $0.6594 \pm 0.0020$ |
| TS = 9 | MSE | $0.0992 \pm 0.0002$ | $0.1862 \pm 0.0002$ | $0.1653 \pm 0.0001$ | $0.1624 \pm 0.0030$ | $0.1665 \pm 0.0020$ | $0.1403 \pm 0.0030$ |
| | L2 | $0.3970 \pm 0.0002$ | $0.7543 \pm 0.0001$ | $0.6408 \pm 0.0001$ | $0.6532 \pm 0.0020$ | $0.6673 \pm 0.0030$ | $0.6622 \pm 0.0010$ |

Table 5: Comparison of benchmarks in terms of MSE and Relative L2 norm on Jellyfish (Fluid).

| Time Step | Metrics | ConvLSTM | ViT | MLP-Mixer | DDOs-FNO | DDOs-SFNO | DDOs-TFNO |
|---|---|---|---|---|---|---|---|
| TS = 1 | MSE | $0.0569 \pm 0.0002$ | $0.2440 \pm 0.0003$ | $0.1841 \pm 0.0001$ | $0.5012 \pm 0.0020$ | $0.5019 \pm 0.0020$ | $0.5563 \pm 0.0010$ |
| | L2 | $0.4719 \pm 0.0002$ | $0.8754 \pm 0.0001$ | $0.8065 \pm 0.0003$ | $0.9019 \pm 0.0020$ | $0.9314 \pm 0.0020$ | $0.9209 \pm 0.0040$ |
| TS = 3 | MSE | $0.0989 \pm 0.0001$ | $0.2608 \pm 0.0003$ | $0.1908 \pm 0.0001$ | $0.5218 \pm 0.0010$ | $0.5367 \pm 0.0020$ | $0.5678 \pm 0.0020$ |
| | L2 | $0.5953 \pm 0.0002$ | $0.9027 \pm 0.0001$ | $0.7217 \pm 0.0001$ | $0.9215 \pm 0.0010$ | $0.9461 \pm 0.0010$ | $0.9287 \pm 0.0030$ |
| TS = 5 | MSE | $0.1428 \pm 0.0002$ | $0.2773 \pm 0.0001$ | $0.2548 \pm 0.0002$ | $0.5517 \pm 0.0030$ | $0.5466 \pm 0.0010$ | $0.5466 \pm 0.0030$ |
| | L2 | $0.6991 \pm 0.0001$ | $0.9247 \pm 0.0002$ | $0.7897 \pm 0.0001$ | $0.9484 \pm 0.0030$ | $0.9566 \pm 0.0010$ | $0.9309 \pm 0.0010$ |
| TS = 7 | MSE | $0.1784 \pm 0.0002$ | $0.2887 \pm 0.0003$ | $0.2756 \pm 0.0001$ | $0.5674 \pm 0.0010$ | $0.5581 \pm 0.0020$ | $0.5522 \pm 0.0030$ |
| | L2 | $0.7792 \pm 0.0002$ | $0.9373 \pm 0.0001$ | $0.8124 \pm 0.0001$ | $0.9518 \pm 0.0010$ | $0.9617 \pm 0.0010$ | $0.9455 \pm 0.0010$ |
| TS = 9 | MSE | $0.2496 \pm 0.0001$ | $0.2965 \pm 0.0001$ | $0.2843 \pm 0.0002$ | $0.5718 \pm 0.0020$ | $0.5718 \pm 0.0020$ | $0.5609 \pm 0.0030$ |
| | L2 | $0.8843 \pm 0.0002$ | $0.9513 \pm 0.0001$ | $0.8249 \pm 0.0001$ | $0.9645 \pm 0.0030$ | $0.9634 \pm 0.0030$ | $0.9534 \pm 0.0030$ |

## A.6 BROADER IMPACTS AND LIMITATIONS

**Broader Impact:** The introduction of comprehensive large-scale datasets, namely Tension, Wets, CellDivision and Jellyfish, has significant implications for the field of biological fluid simulation. These datasets address the challenges faced by the community, including the lack of dynamic biological process capture and limited scale of existing datasets. By providing a standardized evaluation framework and incorporating physical modeling techniques, the datasets empower researchers to objectively assess and compare data-driven methodologies. This fosters advancements in the field and promotes the development of accurate and efficient models for simulating complex fluid dynamics within biological systems. The availability of benchmark datasets also enhances reproducibility and comparability of results across studies, facilitating knowledge sharing and collaboration within the research community.

**Limitations:** While the introduced datasets offer valuable resources for data-driven biological fluid simulation, they may have some limitations. First, the datasets are designed based on specific biological scenarios and may not encompass the full range of biological fluid dynamics. Researchers should be cautious when extrapolating findings to other systems. Second, the datasets rely on physical modeling techniques such as the phase-field method, which may introduce certain simplifications and assumptions that could impact the accuracy and applicability of the results. It is important to consider

the limitations and assumptions of the underlying models when interpreting the data. Finally, the scale and complexity of the datasets may pose computational and resource challenges for researchers with limited access to high-performance computing infrastructure.

