# OpenReview forum: "CellDJBench: Benchmark Datasets for Data-Driven Biological Fluid Simulation"
_ICLR.cc/2025/Conference — Submitted to ICLR 2025_

### Official Review · Reviewer_4bxA · 2024-11-01

**Soundness:** 2
**Presentation:** 2
**Contribution:** 2
**Rating:** 5
**Confidence:** 3

**Summary:**

The authors' work contributes to the biological fluid simulation field. They have identified and addressed key limitations, particularly the lack of dynamic data and the limited dataset scale. To overcome these challenges, the authors have introduced four new large-scale datasets—Tension, Wets, CellDivision, and Jellyfish—which capture dynamic biological fluid scenarios. These datasets, built using physical modeling techniques such as the phase-field method, aim to serve as standardized benchmarks for evaluating data-driven models. The authors' commitment to fostering comparability and reproducibility across studies is a commendable effort that will advance the field, allowing researchers to objectively assess and improve data-driven models for complex biological fluid dynamics.

**Strengths:**

As a non-expert in biological fluid simulations, I find the authors' benchmark to be well-motivated and potentially impactful. It has the potential to benefit not only the biofluid domain but also machine learning researchers interested in applying and developing novel methods for the problems. The aim to create dynamic (time-series) data comparable across experiments is a significant and promising aspect of their work.

**Weaknesses:**

First, no comparison exists to existing benchmarks for solving PDEs and dynamical systems such as [1]. While the scope of the submitted paper is more specific, it still uses simulators based on solving differential equations (or time-evolving dynamics in general). Indeed, the proposed benchmark uses the same neural network surrogate models as prior benchmarks (FNO, U-Net, etc.), or those used recently as surrogate PDE solvers (DPPMs, etc.). So, it is essential to explain why a separate new benchmark is required by contrasting it to existing PDE-based benchmarks.

Second, the benchmark provides only the datasets as a Google Drive link. While this might change after acceptance, there needs to be a discussion on FAIR data principles or how versioning and data availability will be achieved. A benchmark dataset requires versioning, bug fixes, and the possibility of allowing users to file issues and updates. This must be included as part of a dataset/benchmark paper.

Third, existing benchmarks typically include the algorithms used to generate the data. While the paper explains these algorithms, I could not find the code for the solvers included in the repo. Will this be provided to future users? This is important because some users might want to use different parameters, setups, etc.

Fourth, there is a need for a clear explanation of the advantages of the ML methods over the simulators. What exactly is the advantage of the ML methods over the simulators? Are they more efficient? Can they somehow assimilate existing data better? Why do we need them, and what is the minimum bar an ML method needs to meet to be usable and accepted in the community? This wasn’t clear to me, and I believe it's an important aspect that needs to be addressed.

[1] https://arxiv.org/abs/2210.07182

**Questions:**

What exactly is the advantage of the ML methods over the simulators? Are they more efficient, can they somehow assimilate existing data better? Why do we need them, and what is the minimum bar a ML method needs to meet to be usable and accepted in the community? Will the solver/simulator code be provided to users?

---

### Official Review · Reviewer_RvZf · 2024-11-03

**Soundness:** 3
**Presentation:** 3
**Contribution:** 2
**Rating:** 5
**Confidence:** 4

**Summary:**

This is a data set and benchmark paper that focuses on biological fluid simulation. Unlike existing biological data sets, this data set is much larger and aims to capture various important dynamic biological processes. The paper generates and curates four comprehensive datasets: Tension, Wets, CellDivision and Jellyfish. It also presents benchmarking of a number of SOTA SciML techniques based on this data set.

**Strengths:**

1. This is an interesting dataset that captures a wide range of biological fluid dynamics scenarios including encompassing tension-driven flows, wetting phenomena, cell divisions, and the intricate motion of jellyfish-like organisms. These four individual data sets are meticulously curated and incorporates variations in important physical parameters, such as adhesion, tension, and flow rates, to simulate the complexity and heterogeneity of real-world biological systems.

2. The data set contains ~2.5M images and ~130K videos, including both micro and macro scales which should be useful.

3. Most importantly, the authors evaluate 11 data-driven SciML approaches for the biological dynamics modeling problem, based on this data set which should be a useful benchmark for the biological fluids simulations community.

**Weaknesses:**

The actual data samples shown in the paper seem to lack in realism. For example, the cell shapes are rather simple, mostly circular or oval and they do not seem to capture the complexity of most real cells. Without the geometric and morphological diversity of cell samples, the data set may not be rich enough for useful SciML research. The jellyfish dataset also looks quite low fidelity and I am not sure how useful it would be for general applications or for model pretraining.

**Questions:**

1. Can more realistic cell shapes be simulated?

---

### Official Review · Reviewer_MDAV · 2024-11-07

**Soundness:** 4
**Presentation:** 2
**Contribution:** 3
**Rating:** 5
**Confidence:** 4

**Summary:**

This paper provides a large-scale dynamical biological dataset.

**Strengths:**

S1) **Novel Dataset**: A standardized, large-scale dataset capturing dynamic biological processes at both the single-cell and organismal levels.

S2) **Extensive Benchmark**: Comprehensive benchmarking in single-cell biological systems is shown only on paper and is missing in the codebase.

**Weaknesses:**

W1) A more comprehensive comparison of benchmarks is needed. It should include parameters, training time, inference time, and memory consumption.

W2) The experiments require more in-depth analysis and conclusions. Highlighting the top three models would provide valuable insights into their performance.

W3) The complete dataset and models need to be included. As they are part of the datasets and benchmarks track, they should be included here for completeness.

**Questions:**

Q1) **Dataset Availability**: The dataset on Google Drive seems limited. Please provide an open-source version of the complete dataset. Also, I noticed only a few models on GitHub, while the paper mentions 11.

Q2) **Experimental Details**: Could you please elaborate on the training, validation, and testing splits used for the experiments? Additionally, more details on the hyperparameter tuning process for different models and datasets would be helpful.

Q3) **Model Selection**: While it's understandable that testing all models from the literature is impractical, a more detailed discussion on the rationale behind the selected models, or why others were excluded, would be valuable.

Q4) **Open-Source Package**: Would it be feasible to provide the datasets and models as a Python package to facilitate broader community usage?

Q5) **DDPM Training**: How was the DDPM model trained for time-dependent biological dynamics?

Q6) **Repository**: The current repository doesn't align with the claims in the paper. It needs some models and the entire dataset.

Note: **Please ensure that the code must be anonymized as per ICLR guidelines.**

---

### Official Review · Reviewer_2XiY · 2024-11-08

**Soundness:** 3
**Presentation:** 3
**Contribution:** 3
**Rating:** 5
**Confidence:** 4

**Summary:**

The paper presents a new dataset for evaluating the effectiveness of machine learning methods for biological fluid simulation. The new dataset contains 1.3 TB of data from four biology use-cases: Tension, Wets, CellDivision, and Jellyfish. This dataset has been used to compare the performance of operator learning methods such as FNO, SFNO, TFNO, and other models such as DDPM and UNet.

**Strengths:**

1. Important contribution of a new dataset to the ML community in a problem of great scientific relevance
2. Clear explanation of limitations of previous works in comparison with the proposed dataset.
3. Comprehensive evaluation of multiple baseline methods on the dataset with visualizations.

**Weaknesses:**

3. Technical description of the methods used for producing the datasets are not clear and look incomplete. For example, lines 209 to 227 provide a very abstract overview of the inputs and outputs considered in the data-driven biological simulation problem setup. Also, how are the two biological simulation models (phase-field and Lily-pad) connected to the four use-cases (Tension, Wets, CellDivision, and Jellyfish)?4. It appears that the dynamics of organisms such as cells and jellyfish are represented as 2D fields (in the form of imageS). It is not clear whether treating the dynamics of biological objects as fields is valid, when they should rather be treated as discrete objects. A discusison of this potential simplification should be provided.
3. Some datasets and related works are missing such as the extensive literature on cell tracking datasets: https://celltrackingchallenge.net/, https://arxiv.org/abs/2202.04731, https://www.nature.com/articles/s41592-023-01879-y. It will be good to compare the proposed dataset with these existing datasets, and describe the contribution of this paper relative to prior works.
4. The link to the dataset and code is revealing author identity at multiple places such as the username in the Google Drive folder link: https://drive.google.com/drive/folders/1fy5C3RQeIQLk-AM19Zyoo93cy4RZ44D1 and python files with user home directory names (e.g., line 34 of https://anonymous.4open.science/r/BioJCell--9E53/code/tfno_jelly.py). This is violating the double blind submission policy.

**Questions:**

Additional questions:
1. What is challenging or different in biological fluid simulations that is different from other types of fluid dynamics simulations? It will be good to explicitly mention this to establish the contribution of this work beyond just applying ML methods on a new problem.
2. What insights can we draw by analyzing the performance of multiple machine learning models on the proposed dataset? While the comparison results are presented, it will be good to include a discussion of why one method performs better than another on the four datasets, motivating future research in this area.
3. Can this analysis be extended to study multiple cells interacting together?

---

### Meta-Review · Area_Chair_rj9C · 2024-12-22

**Metareview:**

In this work, authors introduce CellDJBench, a comprehensive benchmark dataset for data-driven biological fluid simulation containing four distinct datasets: Tension, Wets, CellDivision, and Jellyfish. The dataset aims to address two key limitations in existing biological datasets: lack of dynamic process capture and limited scale due to experimental constraints. The authors employ physical modeling techniques like phase-field methods to generate 1.3TB of data encompassing various biological fluid dynamics scenarios. They evaluate 11 data-driven approaches including operator learning methods (FNO, SFNO, TFNO) and other architectures (DDPM, UNet) on these datasets.

There are several strengths for the paper as follows. The work addresses a significant gap in biological fluid simulation datasets by providing large-scale dynamic data. Reviewers 2XiY and RvZf noted the comprehensive evaluation of multiple baseline methods with visualizations. The dataset's scale (2.5M images, 130K videos) and coverage of both micro and macro scales represent meaningful contributions to the field. The standardized evaluation framework enables objective assessment and comparison of different methodologies.

However, there are several critical weaknesses as outlined below. Reviewer 2XiY highlights incomplete description of methods and unclear connections between simulation models and use cases. Reviewer RvZf points out that cell shapes appear oversimplified and may not capture real-world complexity raising concerns about the realistic nature of the datasets. Reviewer MDAV's notes on missing models and incomplete dataset availability in the provided repositories raise concerns about reproducibility of the work. Double-blind violations were identified by multiple reviewers in code paths and dataset links. However, this was addressed by the authors. Further concerns were raised regarding weak and incomplete choice of baselines. Reviewer 4bxA's emphasis on the absence of comparisons with existing PDE benchmarks and unclear advantages over traditional simulators.

The implementation and reproducibility issues seriously undermine the paper's core contribution as a benchmark. The incomplete codebase and dataset availability noted by Reviewer MDAV contradict the paper's goal of providing a standardized evaluation framework. For future submissions, the authors should address the technical clarity issues, enhance dataset realism, ensure complete code and data availability, and strengthen comparative analysis with existing benchmarks. The concept has merit but requires substantial refinement to serve as a reliable benchmark for the community.

**Additional Comments On Reviewer Discussion:**

No responses were provided by the authors and hence there were no discussion during rebuttal period.

---

### Decision · Program_Chairs · 2025-01-22

Reject